**Perspective**

# Exploiting neuro-inspired dynamic sparsity for energy-efficient intelligent perception

Sheng Zhou ⬡[1], Chang Gao ⬡[2], Tobi Delbruck ⬡[1], Marian Verhelst ⬡[3] & Shih-Chii Liu ⬡[1] ✉

Artificial intelligence (AI) has made significant strides towards efficient online processing of sensory signals at the edge through the use of deep neural networks with ever-expanding size. However, this trend has brought with it escalating computational costs and energy consumption, which have become major obstacles to the deployment and further upscaling of these models. In this Perspective, we present a neuro-inspired vision to boost the energy efficiency of AI for perception by leveraging brain-like dynamic sparsity. We categorize various forms of dynamic sparsity rooted in data redundancy and discuss potential strategies to enhance and exploit it through algorithm-hardware co-design. Additionally, we explore the technological, architectural, and algorithmic challenges that need to be addressed to fully unlock the potential of dynamic-sparsity-aware neuro-inspired AI for energy-efficient perception.

In response to ever more complex and diverse perception tasks, AI models have grown substantially in both size and computational requirements. This trend follows empirical scaling laws[1], increasing the energy demands for training and inference. It poses a critical challenge to the deployment of AI models, particularly on edge platforms targeting applications such as mobile computing, smart wearables, and autonomous robots, where dynamic real-time interaction with the environment is necessary[2,3].

This Perspective focuses on AI perception systems that process input from sensors of various modalities used for extracting information in natural scenes. These systems typically exploit neural networks consisting of convolutional and recurrent layers, and recently, more complex architectures like transformers. To deploy perception systems on energy-constrained hardware platforms, huge efforts have been made to reduce unnecessary computations within the networks, that is, to increase the compute sparsity, which will improve the energy efficiency of the corresponding hardware platforms.

Traditional approaches focus on what we term static sparsity—sparsifying network connections by applying pruning techniques[4]. To further minimize the size and complexity of the model, pruning is often combined with other optimization techniques such as parameter quantization[5] and neural architecture search[6]. Although these static sparsity methods have yielded substantial model-size reduction and inference acceleration (e.g., 2 × smaller and 1.8 × faster convolutional models for image recognition[7]), these approaches are inherently static and do not account for the characteristics of the actual data input during runtime. Recently, several data-driven dynamic sparsity approaches are on the rise, to further decrease the number of computations at runtime. Yet, this emerging field is still highly scattered, and opportunities for perception systems remain largely underexplored.

This Perspective therefore explores the various forms of dynamic sparsity, with a focus on context-aware sparsity, which seek to reduce computation based on the dynamic structure of the incoming data and the evolving context of a task, particularly for systems that operate in natural environments. This data-driven approach is inspired by the redundancy in the sensor and network output due to intrinsic spatio-temporal correlations of natural stimuli as we will discuss further in the next section. Rather than processing every component of the model for every input sample, a system employing dynamic context-aware sparsity would be selectively activated by the input, and would then execute the network computations and memory accesses only when needed. This concept is inspired by biological brains, which operate under strict energy budgets with tight latency constraints, and have

[1]Institute of Neuroinformatics, University of Zurich and ETH Zurich, Zurich, Switzerland. [2]Delft University of Technology, Delft, the Netherlands. [3]KU Leuven & imec, Leuven, Belgium. ✉e-mail: shih@ini.uzh.ch

evolved to process information in an adaptive, context-dependent manner.

While spiking neural networks (SNNs) operating on data from event-based sensors serve as the prototypical example, we will demonstrate that the concept of dynamic sparsity is much more general and broadly applicable across neural network architectures beyond SNNs. For example, transformers[8,9], the current workhorses of large-scale foundation models, exploit a form of data-driven dynamic attention. Here, the self-attention mechanism takes into account some contextual information from the token sequence. Typical transformers, however, execute this attention mechanism in a dense fashion, primarily towards increased accuracy rather than reduced computation counts. They can also benefit from reduced computation by using the sparse dynamic outputs of event-based vision sensors[10]. However, they leave a lot of margin for further exploiting dynamic sparsity in a data-driven and context-aware fashion, as nature does.

This Perspective outlines the broad potential of dynamic sparsity as a key enabler of the next wave of energy-efficient intelligent perception. We draw on biological insights to demonstrate how the brain exploits dynamic sparsity in various ways, present a taxonomy of the sparsity types, then explore how dynamic sparsity can be introduced at multiple algorithmic and hardware levels through both sparsity-enhancing and sparsity-exploiting techniques. Additionally, we examine open challenges in architectures, algorithms, and technologies, as well as potential applications for future exploration and innovation in dynamic-sparsity-aware, neuro-inspired AI systems. In particular, we focus on the opportunities arising from dynamic sparsity for artificial neural networks (ANNs), where we identify greater potential benefits than for SNNs, which already have closer connections to biology.

### Neural inspiration

Animals can only sustain themselves with the amount of energy they can forage[11], making energy efficiency crucial for survival. Consequently, the brain's computation must be highly energy-efficient. This demand for efficiency suggests that neurons in the brain must fire sparsely, since spike generation accounts for more than 50% of brain energy consumption[12]. Various estimates indicate that the average firing rate of cortical neurons is approximately 1 Hz (Box 1). The sparse firing of neurons can be directly observed in an example calcium imaging recording of brain slices, as shown in Fig. 1A.

The sparsity of neural activity suggests that the brain uses sparse firing patterns to encode information, a concept known as sparse coding[13]. Theoretical and experimental evidence supports this principle across various sensory modalities, including vision[14], audition[15], and olfaction[16]. Sparse coding is consistent with the redundancy-reduction hypothesis[17], which postulates that sensory systems aim to preserve essential information while discarding redundant input. Natural scenes, such as a horse in motion illustrated in Fig. 1B, exhibit high spatiotemporal redundancy: most pixels change little over time, and nearby pixel values are highly correlated. Therefore, encoding only the spatiotemporal changes drastically reduces the number of spikes required to represent the stimuli[17].

Another important property of nervous systems is their statefulness. Neurons maintain localized states through a variety of mechanisms such as synaptic connections, neuron membrane potentials, calcium concentrations, and many other localized, time-varying state variables[18,19]. These states—distributed at different synapses, neurons, and brain areas—allow biological neural networks to integrate sensory information across a range of temporal and spatial scales, forming context-aware models of the environment. This stateful computation approach enables efficient processing: rather than recalculating everything from scratch, the brain updates only what is necessary based on its current state using sparse local communication.

While modern AI models do employ states—such as hidden states in recurrent neural networks (RNNs)[20], KV cache in Transformers[21], and long-term memory banks in memory-augmented models[22]—they typically process all inputs and all model components densely at each inference step. This dense processing undercuts the potential gains from statefulness by incurring high energy and latency costs. In contrast, the brain performs selective and sparse updates, often triggered by surprise or salient stimuli.

Two key mechanisms have been proposed to explain how the brain maintains sparse activity and energy-efficient inference: predictive coding and attention-based gating. Predictive coding[23] posits that the brain actively generates top-down predictions of the incoming stimuli and compares them with the actual inputs. The predictions are then updated by the bottom-up error signals. This feedback process allows the brain to focus its processing resources on unexpected inputs (surprise). For example, in a driving scene (Fig. 1C), the background motion is highly predictable, whereas a child suddenly running across the street generates a significant prediction error, rapidly engaging sensory processing and motor response. Fig. 1D illustrates the consequence of such a predictive model, where the prior established by the first sentence biases the interpretation of "flies" in the second sentence, necessitating a reset. Nevertheless, this bias dynamically lowers the inference cost and latency by constraining the search space.

## BOX 1

# How sparse is the brain's spiking activity?

A dominant form of dynamic sparsity in the brain is the sparsity of the spikes—the fundamental units of neural computation. But how sparse is the brain's spiking activity? More than 50% of mammalian brain power is dedicated to generating spikes[12], and a back-of-the-envelope estimate[a] relating human brain power of $P \approx 10$ W to the average spike rate $R$ suggests that $R \approx 1$ Hz[95,153]. This estimate applies to the entire brain, and neurons in higher cortical areas have much lower spike rates than those near the sensory periphery[13]. If we take the computation time scale to be 1 ms (based on the response time of excitatory postsynaptic potentials), this implies that the human brain's spiking activity is roughly 99.9% sparse, with an active (spiking) duty cycle of only $10^{-3}$. Although the average spike rate is only 1 Hz, we clearly operate with much higher sensing and processing bandwidths.

Since the brain produces spikes only when needed, synaptic operations are likely to be highly significant. This contrasts starkly with current ANNs, which perform multiply-and-accumulate (MAC) operations indiscriminately using oversimplified point neuron models. To make the most of every precious spike, biological neurons employ a series of stateful computations. For example, many biological neurons high-pass filter their input through spike-frequency adaptation[18]. Furthermore, synaptic events have rich dynamics integrated within the nonlinear dendritic trees. Neurons with expansive dendritic nonlinearities and depressing synapses act as novelty filters at a finer scale.

$^a$ $P = 10$ W $= R$ (Hz) $\times 10^{11}$ neurons $\times 10^4$ synapses/neuron $\times (10^{-1}$ V $\times 10^{-10}$ A $\times 10^{-3}$ s)/spike $\Rightarrow R \approx 1$ Hz

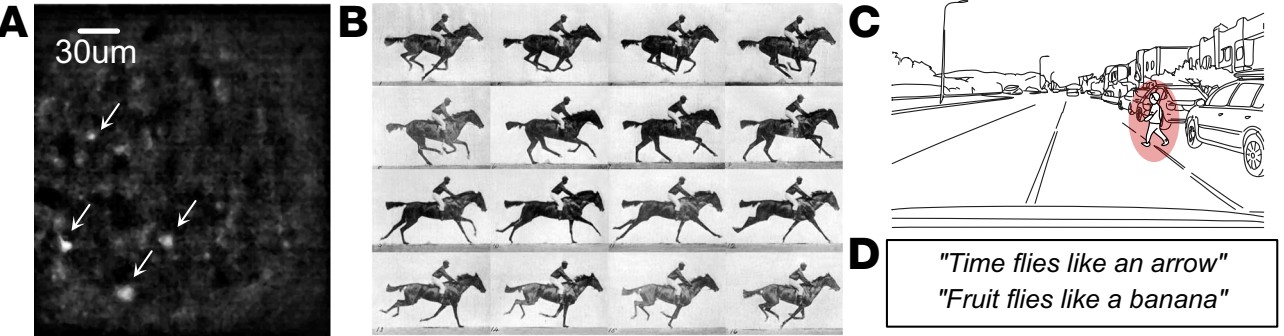

**Fig. 1 | Examples of dynamic sparsity. A** Sparse spiking activity (arrows) observed through calcium imaging of a brain slice from mouse frontal cortex (courtesy R. Loidl). **B** Muybridge's 1878 Horse in Motion sequence repeats nearly exactly the same information across frames, albeit with severe aliasing. **C** Driving sequence is dense and highly repetitive; the critical pixels with the child (circled) are a tiny fraction of total. **D** Example used by J. Hopfield in his Caltech teaching of forming attentional expectation bias in language that sparsifies subsequent inference.

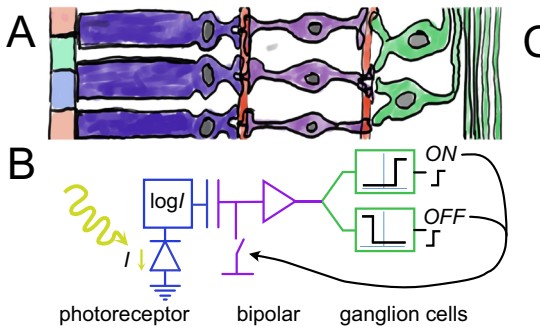

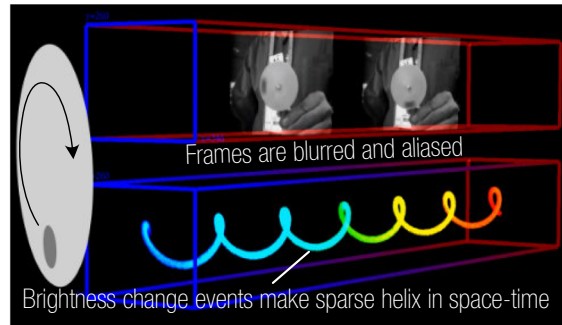

**Fig. 2 | Dynamic sparsity in neuromorphic vision sensors. A** The three layers of the biological retina[25]. Left to right: photoreceptors, bipolar cells, and ganglion cells. **B** Silicon implementation of the retina cells in a neuromorphic event camera pixel[26]. **C** Comparison of dense frames (top) and sparse brightness-change events (bottom) from a spinning dot stimulus, recorded by a hybrid vision sensor[152].

In parallel, attention mechanisms[24] serve as top-down processes that prioritize relevant inputs and modulate the activation of various computational pathways. This form of selective processing constitutes a coarse but powerful implementation of dynamic sparsity. By focusing only on salient information, attention enables the brain to allocate resources more effectively and reduce overall processing cost.

Figure 2 shows an example of embedding neuro-inspired dynamic sparsity in a vision sensor. Retinal circuits respond primarily to changes in the visual field[25], and event camera pixels[26] mimic this behavior by producing output events only when brightness changes above a certain threshold occur. These neuromorphic sensors generate sparse, low-latency event streams that better capture dynamic visual information without the redundancy of frame-based input, offering substantial advantage in terms of latency, temporal resolution, energy efficiency, and dynamic range[27].

The neural foundation of dynamic sparsity as well as its demonstrated effectiveness in neuromorphic vision sensors, motivate the exploration of its broader applications in energy-efficient AI. To connect insights from neuroscience with the recent progress in various fields—such as neuromorphic engineering, deep learning, and domain-specific accelerators—and to systematically frame the key design considerations for implementing this principle, the next section elaborates a necessary taxonomy of dynamic sparsity.

## Types of dynamic sparsity

Sparsity plays a crucial role in both biological and artificial perception systems. By eliminating non-informative redundancy, sparsity reduces unnecessary computation and communication, thereby shortening processing latency and lowering energy consumption. Depending on whether the eliminated redundancy is data-dependent, sparsity in perception systems can be broadly classified into two categories: static sparsity (Fig. 3A) and dynamic sparsity (Fig. 3B).

Static sparsity exploits predetermined and fixed redundancy, resulting in a fixed processing flow during perception. Methods for obtaining static sparsity include fixed duty cycling of sensors[28], using a preset camera region of interest, as well as pruning the weights of a neural network[29]. Although static sparsity effectively reduces computation and data movement demand for a given task, it enforces an identical processing flow regardless of input data variations. This fixed connectivity map can potentially miss out on further data-dependent optimization as discussed next.

Dynamic sparsity, in contrast, leverages data-dependent redundancy. Box 2 provides a formal definition of dynamic sparsity. Our definition of dynamic sparsity is distinct from a class of network pruning methods known as dynamic pruning[30] or dynamic sparse training[31–34]. Although these methods dynamically adjust the sparse neuron connectivity during training, the sparsity is fixed once the training is completed (i.e., during inference). In contrast, we focus on algorithms and hardware designs targeting sparse computational flow that can dynamically change in a data-driven fashion during inference.

Prior works that have discussed and incorporated various forms of dynamic sparsity are often applied to solve specific, isolated problems, resulting in a fragmented landscape. For example, some works focus exclusively on activation sparsity in convolutional neural networks (CNNs) (e.g., skipping zero-valued ReLU outputs[35,36], dynamic channel and activation pruning during inference[37]) or subnetwork gating for large language models (LLM) (e.g., Mixture of Experts (MoE)[38–40], and speculative decoding[41–43]), while others explore stateful temporal sparsity in RNNs (e.g., delta networks[44,45]). These various forms of dynamic sparsity have rarely been analyzed within a unified

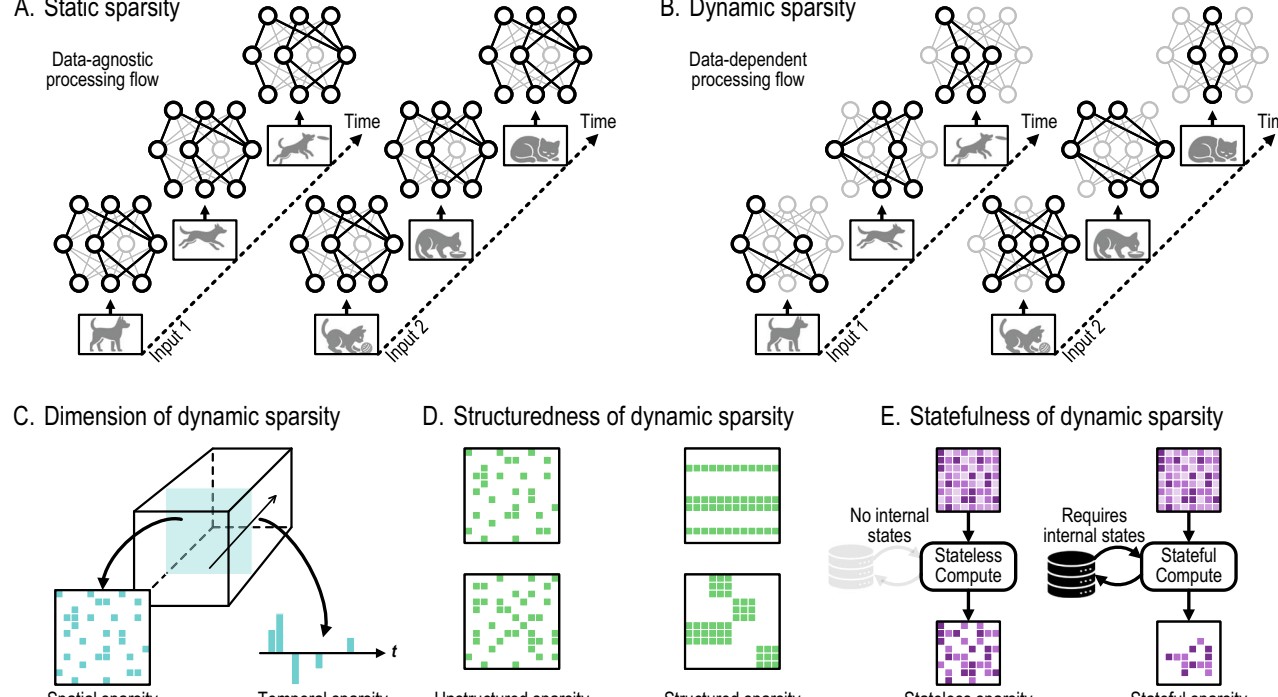

**Fig. 3 | Taxonomy of sparsity. Sparsity is classified based on whether it is data-dependent. A** Static sparsity is fixed and leads to a static processing flow. Weight sparsity, commonly employed in neural network compression, falls into this category. **B** Dynamic sparsity is data-dependent and leads to a dynamic processing flow. Rooted in data redundancy, it can be further categorized based on its dimension, structuredness, and statefulness. **C** Dynamic sparsity can be spatial, temporal, or spatiotemporal, depending on the dimension along which the information redundancy is exploited. **D** Dynamic sparsity can be either structured or unstructured, depending on whether such sparsity should satisfy any spatial or temporal structural constraints. **E** Dynamic sparsity can be either stateless or stateful, depending on whether extra memory or states are employed to induce sparse representations from dense representations.

## BOX 2

# A formal definition and taxonomy of dynamic sparsity

We model the computation of an AI system as a (possibly stateful) mapping $\Phi : \mathcal{X} \times \mathcal{S} \to \mathcal{Y} \times \mathcal{S}$, where $\mathcal{X}$ is the input space (e.g., sensory inputs), $\mathcal{Y}$ is the output space (e.g., predictions or control outputs), and $\mathcal{S}$ is the state space (e.g., recurrent states or internal memory). For any $(\mathbf{x}_t, \mathbf{s}_t) \in \mathcal{X} \times \mathcal{S}$, where $\mathbf{x}_t \in \mathcal{X}$ is the current input and $\mathbf{s}_t \in \mathcal{S}$ is the current state, we write $\Phi(\mathbf{x}_t, \mathbf{s}_t) = (\mathbf{y}_t, \mathbf{s}_{t+1})$, where $\mathbf{y}_t \in \mathcal{Y}$ is the current output and $\mathbf{s}_{t+1} \in \mathcal{S}$ is the updated state.

We index the operations to compute $\Phi$ by $i = 1, 2, ..., n$, where $n$ is the total number of operations. Each operation might be low-level (e.g., scalar multiplications or additions) or high-level (e.g., activation of a sensor or a sub-network). To sparsify $\Phi$, we introduce a (possibly time-dependent) binary mask $\mathbf{m}_t = (\mathbf{m}_{t,1}, \mathbf{m}_{t,2}, ..., \mathbf{m}_{t,n}) \in \{0, 1\}^n$, where $\mathbf{m}_{t,i} = 1$ means operation $i$ is executed when processing $(\mathbf{x}_t, \mathbf{s}_t)$, and $\mathbf{m}_{t,i} = 0$ means operation $i$ is skipped. Thus, when using the sparsified mapping (denoted as $\Phi_{\mathbf{m}}$) to compute $(\mathbf{y}_t, \mathbf{s}_{t+1})$, only the operations with $\mathbf{m}_{t,i} = 1$ are performed.

The sparsity of $\Phi_{\mathbf{m}}$ is static if the mask $\mathbf{m}$ is fixed during inference (e.g., obtained by offline pruning) and does not depend on $\mathbf{x}_t$ or $\mathbf{s}_t$. In

contrast, the sparsity is dynamic if $\mathbf{m}$ is determined on-the-fly based on $\mathbf{x}_t$, $\mathbf{s}_t$, or both. In other words, $\mathbf{m}_t = g(\mathbf{x}_t, \mathbf{s}_t)$ is a function of $\mathbf{x}_t$ and $\mathbf{s}_t$.

With this framework, we can also formalize the proposed taxonomy of dynamic sparsity:

1. Sparsity dimension (Fig. 3C):
   - Spatial: For a given time $t$ and a set of operations $\mathcal{I}$, $\mathbf{m}_{t,i} = 0$ for some $i \in \mathcal{I}$.
   - Temporal: For a given operation $i$, $\mathbf{m}_{t,i} = 0$ at certain time $t$.

2. Structuredness (Fig. 3D):
   - Unstructured: $\mathbf{m}_t$ can take any pattern in $\{0, 1\}^n$.
   - Structure: $\mathbf{m}_t$ is restricted to a subset of patterns $\mathcal{P} \subset \{0, 1\}^n$.

3. Statefulness (Fig. 3E):
   - Stateless: $\mathbf{m}_t$ depends only on the current input $\mathbf{x}_t$ but not on the state $\mathbf{s}_t$.
   - Stateful: $\mathbf{m}_t$ depends on state $\mathbf{s}_t$, and thus also on the past inputs $\mathbf{x}_1, \mathbf{x}_2, ..., \mathbf{x}_{t-1}$.

framework. While existing surveys on dynamic neural networks[46] or ephemeral sparsification[29] summarized the algorithmic aspects of dynamic sparsity within neural networks, a systematic treatment of dynamic sparsity for intelligent perception systems, encompassing both algorithm design and hardware optimization throughout the entire processing chain, is still missing.

As a first step towards a more unified view and to encourage a more holistic approach to system design, we categorize dynamic sparsity along three independent yet interrelated aspects: sparsity dimension, structuredness, and statefulness. This taxonomy of dynamic sparsity is applicable throughout the perception pipeline, from the sensory periphery and early feature extraction to multi-

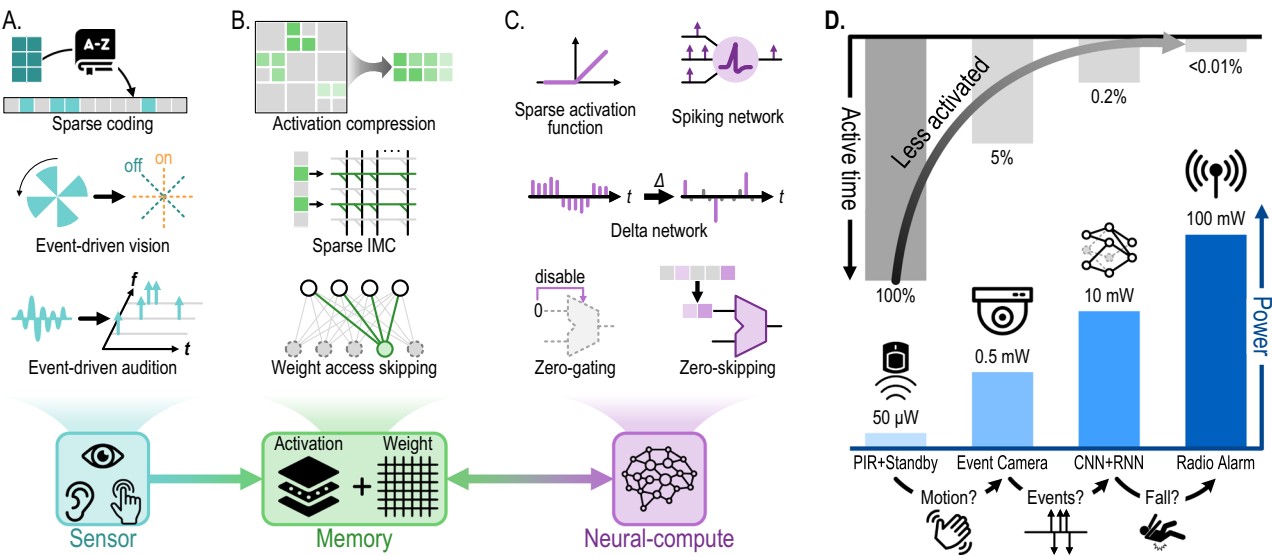

**Fig. 4 | Enhancing and exploiting dynamic sparsity in perception systems.**
**A** The sensor subsystem uses methods such as sparse coding and event-based representation to suppress data redundancy at the very first stage of perception. **B** The memory subsystem provides storage for both activations and weights. It exploits dynamic sparsity by reducing the data traffic to and from the memory, using methods such as activation compression, sparse in-memory computing (IMC), and weight access skipping. **C** The neural-compute subsystem enhances dynamic sparsity through both stateless (e.g., ReLU) and stateful approaches (e.g., spiking network and delta network). Techniques such as zero-gating and zero-skipping exploit the induced sparsity. **D** At the system level, dynamic sparsity brings further energy savings by de-activating and gating entire system modules.

modal integration towards higher-level decision-making. The taxonomy presented here is based on pixel- or neuron-level dynamic sparsity, and we extend it later to a coarser granularity when discussing system-level dynamic sparsity.

### Dimension of dynamic sparsity
Spatial sparsity (Fig. 3C, left) refers to the sparse activity of a collection of neurons or pixels within a time window. It originates from information redundancy along the spatial dimensions. Examples of spatial redundancy are zero-values in the feature maps in CNNs[35,36], the sparsely firing channels/pixels/taxels of event-driven neuromorphic sensors[26,47,48], and the similarity between spatially neighboring pixels[49] or neurons[50] at a given time point.

Temporal sparsity (Fig. 3C, right) refers to the sparse activity of a single neuron or pixel over time. It takes advantage of information redundancy in the temporal dimension. Examples of temporal redundancy are the predominance of environmental noise for speech processing tasks[51], the spectral similarity between neighboring audio frames[52], the slow variation of neuron activation over time[44,45], and the dynamically gated neuron updates in RNNs[53,54].

Spatial and temporal dynamic sparsity are not mutually exclusive. In fact, many stimuli exhibit redundancy in both space and time, leading to spatiotemporal sparsity. For example, in the driving scene shown in Fig. 1C, the relevant objects, such as vehicles and pedestrians, are normally located in the bottom half of the camera view, while the top half can be mostly regarded as background and ignored, exhibiting spatial sparsity. Meanwhile, the movement of the vehicles or the traffic lanes are highly predictable, exhibiting temporal sparsity. Spatiotemporal sparsity can be directly visualized in Fig. 2C, where the sparse brightness-change events create a helix in spacetime.

### Structuredness of dynamic sparsity
Unstructured sparsity (Fig. 3D, left) allows for arbitrary patterns of inactive neurons. There is no restriction as to which neurons can be active or inactive at any moment. Many neuromorphic sensors[26,47,48] as well as spiking[55,56] and non-spiking[35,57] neural network accelerators, utilize unstructured sparsity. Without structural constraints, it

provides the finest sparsity granularity and maximum flexibility in skipping useless computations.

Structured sparsity (Fig. 3D, right), on the other hand, requires the sparse elements to have regular patterns. In general, this entails grouping the neurons so that those within the same group are all active or inactive simultaneously. The neuron grouping defines the granularity of structured sparsity. Example groupings are locally neighboring elements[58], entire rows or columns[59], CNN feature maps[60], and all neurons in the same layer[61]. Such regularity allows for more efficient hardware implementations compared to unstructured sparsity.

### Statefulness of dynamic sparsity
Stateless sparsity (Fig. 3E, left) does not require any internal states to induce sparse representations from dense representations. It relies solely on the instantaneous input to identify redundant operations and determine the sparse computational pattern. Skipping zero activation values in a neural network[35,36] provides a canonical example of stateless sparsity.

Stateful sparsity (Fig. 3E, right) derives the sparse representation by taking into account not only the current input but also an internal state variable that encodes the past inputs. Examples that incorporate stateful sparsity are spiking neuron models implemented in neuromorphic spiking sensors[26,47,48] and SNN processors[55,56]. Notably, the highly sparse computation in the brain is inherently stateful due to its complex dynamics, suggesting the potential advantage of stateful sparsity over stateless sparsity.

### Dynamic sparsity enhancing and exploitation techniques
The brain's ability to induce and exploit dynamic sparsity has long inspired designers of intelligent perception systems, be it robots, wearables, or smart spaces. In this section, we review these state-of-the-art techniques in light of the proposed taxonomy and identify the key design considerations for leveraging dynamic sparsity. As shown in Fig. 4, dynamic sparsity can be incorporated within the three major components of an intelligent perception system, namely, the sensor, memory, and neural-compute subsystems. In addition, it can also be

applied at the system level, which involves dynamically activating the entire modules or subsystems.

## Sensor subsystem

Exploiting dynamic sparsity at the sensor subsystem—the very first stage of the processing pipeline—offers a significant advantage in terms of system-level energy and latency, as much less sensory data needs to be transmitted or processed by the subsequent stages[62]. Both stateful and stateless techniques can be applied to substantially improve energy efficiency and reduce the burden on later processing stages.

The most widely used stateless methods for initial sparsification include sparse coding and vector symbolic architecture (VSA) (also known as hyperdimensional computing). Sparse coding aims to represent input signals using an overcomplete set of basis vectors, ensuring that only a few coefficients are nonzero, so the data representation is highly sparse[13]. Similarly, VSA employs high-dimensional sparse vectors to encode information, naturally promoting sparsity in the activation space by emphasizing zero-valued components in representations[63]. While these stateless techniques effectively exploit the instantaneous sparsity of the original signal, they are inherently limited in exploiting temporal correlation as they lack internal states to memorize previously seen input patterns.

Stateful methods can be employed to achieve higher sparsity levels. These methods leverage past information or spatial correlations to encode the data more efficiently. Compared to stateless methods, stateful methods are particularly effective in natural environments where input signals have strong spatiotemporal correlations, because they dynamically adapt to the characteristics of the signals. Using these correlations, algorithms can drastically reduce power consumption and bandwidth requirements, making them ideal for resource-constrained perception tasks.

A prominent example, shown in Fig. 2, is the neuromorphic dynamic vision sensor (DVS)[26], also known as the event camera[27]. In DVS, pixels use delta modulation[64] to remove temporal redundancy by asynchronously quantizing temporal changes in scene brightness (the logarithm of intensity) as ternary ON/OFF events that encode the location, time, and brightness-change polarity. After each event, the current brightness is stored in the pixel on a capacitor to detect the next change. The sparse event-based camera output enables the subsequent neural network to selectively process only reflectance changes on an event-by-event[65] or patch-by-patch basis[10]. A simple scheme, such as processing accumulated event frames only when event counts reach a few thousand, can effectively save idle computation without compromising latency[66]. To further increase the output sparsity, spatial filtering before the temporal delta modulation removes spatial redundancy[49]. Although maintaining the states requires extra circuit area and energy, the resulting enhancement in output sparsity can reduce the response latency to sub-millisecond under most illumination conditions[26], sensor output bandwidth by more than 100×[67] and the computational burden on subsequent stages by 20×[65] compared to frame-based cameras. Advances in image sensor wafer stacking have reduced the complex pixel size to only a few times that of standard frame-based imagers[68].

Another example of a stateful spiking sensor is the neuromorphic silicon cochlea[47], which uses leaky integrate and-fire (LIF) neurons to generate sparse outputs. Specifically, a LIF neuron maintains a state using its membrane potential that integrates the input current. When the integrated value crosses a threshold, an output pulse is generated, and the state is reset. Therefore, the amplitude of a constant input is converted into a corresponding output pulse frequency, naturally producing a sparser output for a low-amplitude input[69]. However, using a LIF neuron to encode an input sound can lead to a large number of events unless the input sound is largely absent. To address this, the silicon cochlea leverages the time-varying temporal frequency composition of natural sounds by filtering the original sound through different frequency channels[70] before applying event-based encoding. The resulting output events are sparse across both frequency channels and time. This event readout can reduce computational cost by 40×[47] and achieve better localization accuracy for short latencies below 500 ms compared to generalized cross-correlation algorithms[71].

The fundamental consideration in designing dynamic-sparsity-aware sensor subsystems is the trade-off between the cost of inducing dynamic sparsity and the gain from exploiting it. The costs include larger pixels, potential information loss, extra encoding/decoding circuits, and state maintenance overhead for stateful methods. The gains include energy, bandwidth, and latency savings in sensor readout and subsequent processing. This trade-off can be addressed in three ways: (1) Reducing the cost of inducing dynamic sparsity, such as sharing periphery circuits via time-multiplexing[72], imposing structural regularity on the sparsity[58], devising more power- and area-efficient circuits[73] and leveraging advanced fabrication technologies[68]. (2) Improving the gain of exploiting dynamic sparsity, such as conditioning the signal to enhance sparsity[49,70], applying power- and clock-gating to idle circuits[52] and skipping incoming events whenever possible[10,67]. (3) Striking a balance between cost and gain, which involves analyzing the data distribution for the targeted application and selecting the most appropriate implementation, as demonstrated in ref. 74 for voice activity detection.

## Memory subsystem

The memory subsystem is a critical bottleneck in modern computing systems, consuming a significant portion of the system's area and energy footprint[75,76]. As such, the design of the memory subsystem of an intelligent perception system must be meticulously planned to fully leverage the benefits of dynamic sparsity. In the context of brain-inspired computing, memory is primarily used to store weights and activations, which have distinct requirements in terms of memory size, latency, and bandwidth. The difference between the less frequently updated weight memory and the activation memory necessitates unique design trade-offs when exploiting dynamic sparsity.

The activation memory is a natural candidate where dynamic sparsity can be exploited. In biological neural networks, the action potentials are transmitted from one neuron to another through axons, requiring no additional storage or buffering for the sparse neural activities. While implementing such a direct routing scheme in hardware is possible for tiny neural networks[77,78], routing congestion and energy overhead dominate as the network grows larger and more complicated[79]. Eventually, this approach becomes infeasible with current technology.

In modern neural processing systems, this routing issue is resolved by buffering the intermediate activations in memory after computation and loading them from memory later. By leveraging dynamic sparsity, the activation data can be compressed before writing to the activation memory, thereby reducing the required memory capacity and bandwidth. Depending on the characteristics of the activation sparsity, different encoding methods and compression algorithms can be applied. Stateless methods, such as sparsity map[35], run-length encoding[80,81], Huffman coding[82], and least-square fitting[83], are relatively simple to implement and can achieve a moderate compression ratio of up to 5×[35]. Stateful methods, such as bit-plane compression[84] or feature-map-based compression[85], leverage extra state memory to achieve a compression ratio exceeding 10×.

Although the weights are static, the weight memory can also leverage and benefit from the dynamic sparsity of the activations. In the biological brain, the synaptic weights are co-located with the neurons. Such an organization allows computation to be performed without expending energy or time to move the weights to the compute unit. To achieve the same goal in hardware, similar weight memory organization can be implemented using in-memory computing (IMC)

architectures[76,86,87]. In these architectures, the weights are stored in a matrix of memory cells capable of performing MAC operations in-place. The input activations are sent through the bit lines along the rows, and the output activations are obtained from the word lines along the columns. While exploiting weight sparsity in IMC architectures is a challenge[88], dynamic activation sparsity more easily offers energy and latency savings to IMC by reducing the frequency of bit line activations. For example,[89] showed that when there are many zeros in the activation, more bit lines can be activated simultaneously to reduce compute latency by up to 2.7× for typical CNN models. By using bit-serial encoding for input activations, savings can even be achieved when the activation magnitudes are small but not exactly zero, as demonstrated in ref. 90 for diffusion models. Similarly, by using binary events to encode input activations, IMC accelerators for SNNs[77,78] naturally scale their power consumption with the input activity rate.

Although IMC architectures closely mimic the organization of the neural system, scaling them to larger networks[91,92] using current complementary metal-oxide semiconductor (CMOS) technology is extremely costly. Assuming 1-bit precision, storing all $10^{15}$ synaptic weights of the human brain in a 2 nm CMOS process[93] would require 26 m$^2$ of chip area, which is four orders of magnitude larger than a typical die[94]. Therefore, in many systems, the network weights are stored in dedicated memory with higher density, such as off-chip DRAM, and must be moved to the arithmetic units before computation[95]. Since accessing off-chip DRAM requires 100× more energy while providing less than 0.01× bandwidth compared to on-chip SRAM[75], dynamic activation sparsity can offer a huge power and latency advantage by skipping all DRAM access for the fan-out weights of an inactive neuron. For example,[96] achieved a 10× speedup with 90% temporal activation sparsity for an RNN, while[97] reduced the generation latency by 1.8× with 50% sparsity for a transformer-based LLM.

Just as in the sensor subsystem design, it is crucial in memory subsystem design to balance the hardware overhead of handling dynamic sparsity with the resulting energy and bandwidth savings. Unlike sparse weights, which can be statically compressed before deployment, sparse activations must be handled on-the-fly. This overhead can be controlled through either dedicated hardware encoders/compressors in the memory interface/arithmetic unit front-end or through optimized software-level implementations. For instance, on programmable accelerators like GPUs, dedicated GPU kernels could be used to manage dynamic sparsity by optimizing dataflow without requiring specialized circuitry[57,98]. On application-specific hardware, designers often use dedicated silicon area to minimize the sparsification and encoding overhead to dig out as much speedup as possible from the activation sparsity[35,99].

Often, techniques with more aggressive data compression rates come with more complex encoder/decoder designs and increased hardware overhead. This trade-off can be addressed in two ways: (1) Avoiding the overhead of irregular memory access by aiming for structured instead of unstructured dynamic sparsity. This enables the memory subsystem to fetch and store a fixed amount of data words per compressed data tile and maintain data layout regularity[100,101]. While structured sparsity[102,103] has seen adoption in commercial products for static weight sparsity, it is still under-explored for dynamic sparsity. (2) Dynamically adapting the compression mode and sparsity handling method according to the specific levels of dynamic sparsity[104]. This allows for dynamically switching between optimized configurations based on the data statistics.

### Neural-compute subsystem

Electronic systems with real-time heterogeneous sensory input increasingly process these signals using neural networks. The neural-compute subsystem is, hence, another crucial area where dynamic sparsity can be exploited to reduce computation and improve energy efficiency.

Stateless dynamic sparsity in the intermediate activations of neural networks naturally arises from sparse activation functions such as ReLU[105], thresholded ReLU[106], and Sparsemax[107]. Training methods such as $L_1$-regularization[108] that penalize large activation values further enhance the dynamic sparsity. By applying magnitude-based sorting and thresholding, sparsity can also be induced for other activation functions that do not produce zero-valued outputs, such as softmax[38] and sigmoid[54]. While sparse activation functions produce unstructured dynamic sparsity with an unpredictable sparsity level, the sorting-based methods[38,54] lead to structured dynamic sparsity since the number of active neurons in each layer is always fixed. Similarly, the winner-take-all mechanism[109] also enforces structuredness in sparsity by retaining only the largest activation. This structuredness results in more predictable workloads, which are easier to exploit at the hardware level.

The dynamic sparsity introduced by various algorithms can be exploited by hardware MAC units using features such as zero-gating and zero-skipping. With zero-gating, the MAC units are dynamically deactivated to reduce dynamic power upon encountering zero-valued operands. With the levels of sparsity observed in typical CNNs, a 1.6× energy saving can be achieved[36]. Compared to zero-gating, zero-skipping allows more gains by processing only non-zero values through data-dependent scheduling. This improves the utilization of MAC units and leads to additional 2.3× energy savings compared to zero-gating[81]. Stateless dynamic sparsity is gradually being adopted in production-scale models[110] and accelerators[111].

The prototypical example of statefully sparse neural networks is SNNs, in which each neuron maintains its membrane potential as a state and emits spikes only when its membrane exceeds a threshold[112,113]. Various SNN accelerator designs have been proposed to verify the feasibility of this neuromorphic computation model[55,56,114]. Yet, other networks can also exploit stateful sparsity with less of the major SNN drawback of unpredictable memory access. In delta networks[44], fully connected RNNs retain their previous neuron activation as states and compute new activation only if the activation change exceeds a threshold, thus inducing dynamic sparsity at the column level of the weight matrix. CNNs[115] and transformers[116] can undergo a delta transformation so that a neuron holds state in return for fewer operations but more memory for holding state. Also, LLMs use state, in the form of the KV cache to avoid re-computation of the data elements[117,118], and smart KV caching optimization techniques try to reduce this state while maintaining state information[119–121]. Taking this one step further, the state can be more than just the previous activation value. By equipping each neuron with an additional gating input that determines when the neuron is allowed to communicate its output[53], the neurons can be activated more intelligently using a combination of spatiotemporal information.

Dynamic sparsity benefits are not for free. Gating or skipping of redundant computations are accompanied by overheads in control, memory, and scheduling that demand analysis. For example, the control and scheduling overhead differ greatly between unstructured and structured sparsity. Unstructured sparsity[122] creates irregular, data-dependent memory access patterns, complicating hardware schedulers, which must dynamically generate addresses for non-zero data, introducing latency and causing significant workload imbalances across parallel processing elements. One way to address this is to design intrinsically structured dynamic sparsity, e.g., delta networks[44], process entire weight matrix columns corresponding to above-threshold activation vector changes, dramatically simplifying control logic and ensuring predictable, regular data access. This regular sparsity structure can also be imposed by dropping a fraction of activation values[123,124]. Moreover, run-time load balancing can also be used through sparsity-dependent input and output data rerouting, as seen in SpArch[125].

Stateful sparsity also increases memory requirements, which can overshadow the benefits. A delta network, for example, already stores its hidden states, but it must also store the previous pre-activation state of each neuron. However, since the state space of RNNs is tiny compared to the weight space, a recurrent layer with 512 neurons with 16-bit precision requires only an additional 1 kB. But for feedforward CNNs, using temporal sparsity may not make sense because the state spaces (feature maps) are often larger than the number of weights. Using temporal sparsity with these architectures requires holding all units in memory all the time, and reading feature maps to check for changes before writing new values. For these architectures, temporal sparsity might only benefit very sparse applications like surveillance[115].

When applying stateful sparsity in large models, this state footprint can become prohibitive unless mitigated by new techniques that can compress the state itself or recompute it on the fly when needed. Overall, balancing the complexity of hardware implementations with the benefits of sparsity remains a critical challenge.

### Modular system-level dynamic sparsity

The subsections above focus mainly on the exploitation of fine-grained dynamic sparsity in individual subsystem components, such as within single accelerator cores. SoCs increasingly exploit dynamic sparsity across the three subsystem levels to improve energy efficiency and latency. Examples include designs that do keyword spotting[126,127] and face recognition[128,129]. In these state-of-the-art designs, spatial and temporal sparsity are leveraged at the level of system modules, in which complete subsystems are dynamically activated and de-activated during system operation. More examples can be found in consumer mobile electronics, implanted biomedical devices, and space missions[130], which must run on limited energy. These systems use wake-up sensors to monitor the information content of incoming sensor data, and only selectively wake up other system components when deemed useful and necessary.

Figure 4D illustrates such exploitation of dynamic sparsity at the module level, in the context of a hypothetical intelligent sensor that detects elderly fall accidents[131]. At the lowest power level, only the passive infrared (PIR) motion detector is on while everything else is sleeping, and the standby power is on the order of 50 µW[132]. A motion event detected by the PIR sensor turns on a sub-milliwatt event camera[133]. Its sparse output with activity-dependent event rate drives a small CNN that detects the locations of human joints[134]. The input frames and layer activities are extremely sparse, and the CNN hardware exploits dynamic sparsity to skip nonzero activations. The resulting low-dimensional joint position locations then drive a small RNN using temporal sparsity to skip operations[45]. Together, these neural networks burn about 10 mW[129,135,136] but are active only 0.2% of the time. Only when the spatiotemporal pattern of joint motions indicates a fall, will the radio (around 100 mW) be briefly turned on to alert caregivers. But this radio transmission occurs so rarely that the average power is kept below 100 µW, allowing continuous operation on a small battery for years.

In large-scale generative AI, similar hierarchical and modular activation strategies also start to emerge. In speculative decoding[137], a small draft model proposes token sequences that are then selectively verified by a larger target model, thereby gating compute in a data-driven way. MoE[38–40], on the other hand, is a technique where only a subset of specialized sub-networks (experts) are activated for each input, allowing the model capacity to scale efficiently without increasing computation for every input. A gating network decides which experts to use, enabling dynamic routing and improving both accuracy and efficiency. Similarly, recent studies[110,138] demonstrate that activation sparsity can be exploited within LLMs—particularly recurrent ones—to reduce inference energy without loss in accuracy. These techniques reflect a growing interest in applying dynamic sparsity principles at architectural and algorithmic levels in mainstream AI.

## Outlook

Dynamic sparsity, especially context-aware or task-aware sparsity, holds great potential in improving the energy efficiency of perception systems that operate in natural environments. In addition, real-world signals recorded in naturalistic interactions can be statistically sparse, thereby offering computational benefits for dynamic-sparsity-aware systems. For example, a 3.5-day overhead activity-driven event camera recording of a mouse in its cage is over 60,000 × smaller than a 1 kHz monochrome camera recording with the same spatiotemporal resolution[71]. The reduced sensor data leads directly to reduction in computes within the postprocessing network.

By adopting sparsity-enhancing techniques described earlier, the neural networks will further provide more dynamic sparsity. As demonstrated using a delta network[136], we measured 67% dynamic sparsity using a spoken language understanding benchmark[139], representing a modest 3 × savings. We further tested the same system on a 24 h working-day cellphone audio recording from one of the authors. The phone was mostly on the person except during sleeping hours. For this recording of normal everyday sounds, the average dynamic sparsity in the network was over 95%, representing a 20 × savings.

Today's solutions for perception systems still do not go far enough in terms of brain-inspired stateful dynamic sparsity. Current AI perception systems—such as vision systems—typically use stateless networks that require a full network update for each input frame, independent of the computed information from the past. Dynamic sparsity is still beneficial for these networks when deployed on hardware that supports the sparsity type, e.g., zero-skipping in CNNs[35,36]. Accelerators employing stateless dynamic sparsity have already entered mass production[111]. Stateless methods require minimal shift in both neural network and hardware architecture, and therefore will bring advantage in the short term.

In the long term, however, we expect stateful dynamic sparsity to hold more potential because it can exploit the context encoded in the states (Box 2) with a closer connection to dynamical biological networks. To fully unlock the potential of neuro-inspired dynamic sparsity, we believe it is crucial to investigate how states can be used to further enhance the sparsity level (Table 1), especially when we move to networks that use information from multiple sensors and solve more complex tasks. For example, for object detection and tracking, we can leverage context-aware sparsity so that a complete update of the network is not needed for each incoming frame. Therefore, we have to push further along several axes to enable the multi-sensory systems of the future.

We further elucidate the role of dynamic sparsity in a stateless system versus a stateful system in Fig. 5. In a stateless system (Fig. 5A), the layers in the hierarchy are updated sequentially in time for each input frame. Dynamic sparsity-enhancing techniques described earlier can be included in the system modules to reduce the number of computes and memory fetches. Going one step further, adding states to a neuron along with local recurrence as in SNNs or RNNs (Fig. 5B), can help to further sparsify the signal output.

Finally, including top-down feedback (Fig. 5B) through bidirectional connections introduces some form of speculative operation into the overall system, enabling it to dynamically activate only certain modules at the lower level or a sub-network within them. Yet, the amount of sparsity will strongly depend on the information encoded in the states: the better the system can predict the next incoming signal based on its current states, the more computes can be saved. This form of context-aware sparsity can be useful for networks that, for example, are trained to attend to a specific object in a scene. Likewise, biological systems spend more attention (computational power) upon unexpected events using attention mechanisms and predictive models in brain computation. Hence, the outputs, only need to carry the prediction error, as proposed in various neuroscience literature[140,141]. This would be possible if the stateful systems become self-learning systems,

**Table 1 | Limitations and opportunities for dynamically sparse perception systems**

| Current limitations | Bio-inspired opportunities |
|---|---|
| Significantly increasing sensor modalities and data volumes is challenging in perception systems, even when stateless sparsity is exploited. | Stateful techniques allow to further boost sparsity due to the high spatiotemporal correlations present in sensory inputs. |
| Most state-of-the-art stateful sparsity-aware hardware utilizes very simplistic notions of state, such as (a linear combination of) a neuron's past inputs. | Advanced states can pursue the prediction of the expected inputs, instead of mimicking the past inputs. This allows updates only upon surprise. |
| Today's stateful systems compute, retain, and utilize state purely within one computational building block (e.g., a neuron or neural layer), raising the cost of its computation, limiting its predictive value, and exploitation opportunities. | State information should be shared between different system components, and especially be fed back from higher intelligence to lower sensory layers, just like the brain feeds expectation values back into the lower areas of the cortex. |
| Stateful dynamically sparse systems lack a proper hierarchical organization, and do not exploit states at and across multiple abstraction layers towards dynamic activity gating. | Stateful dynamic sparsity should be formalized and unified across abstraction layers, to allow a coordinated dynamic (de)activation of blocks with different granularity. |

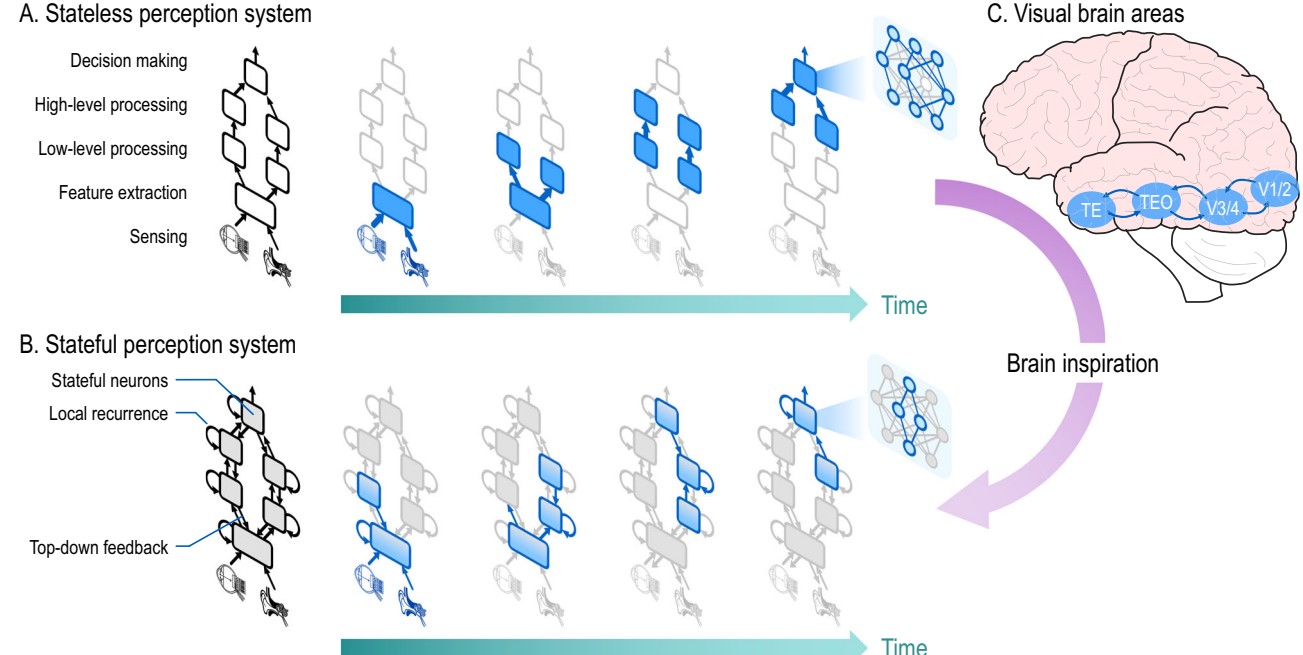

**Fig. 5 | Brain-inspired perception with stateful dynamic sparsity.** Thinner arrows indicate sparser data flow. Boxes filled in grey indicate stateful modules. Boxes filled in blue are activated modules, where lighter fill color indicates fewer activated neurons. **A** Hierarchical updating of the modules across time for a stateless system. Neurons within each module are sparsely activated (see Fig. 3). **B** More dynamic sparsity enabled through stateful systems. Bidirectional connections across modules indicate the bottom-up feedforward and top-down feedback seen in many brain areas. **C** Early visual areas V1/V2 extract low-level features, higher-level areas V3/V4 extract more complex features, and inferior temporal areas TEO/TE are involved in visual processing and object recognition. The top-down feedback helps further reduce the signal transmission between the modules.

capable of learning the patterns in the processed information. As shown in Fig. 5C, the early visual areas in the brain are mutually connected with the higher-level visual areas, which in turn are connected to two key areas of the inferior temporal cortex responsible for visual processing and object recognition. The mutual connections allow both feedforward and feedback processing.

The realization of the vision projected here can, however, quickly become too expensive in terms of the computation of such advanced states, at the risk of introducing more overheads than the potential savings. This will likely be the case when the computation and exploitation of states is left to a single compute entity (a neuron or a neural layer). The overhead can only be kept under control if states are computed and shared between a larger set of entities. This has two consequences. Firstly, it will require communication between different hierarchical layers, with especially the introduction of a feedback path from higher abstraction layers towards the lower sensory layers. Secondly, taking this one step further, this calls for a unified theory and approach for stateful dynamical system (de)activation across different hierarchical layers of abstraction.

## Research directions

To enable the envisioned advanced dynamically sparse perception systems of the future, the following research directions shown in Fig. 6, should be explored further, from neuroscience to device technology:

1. At the neuroscience level, a better understanding of the mechanisms used by brains to dynamically sparsify neural activity is needed, for example, through concepts of attention, saliency, working memory[142], and learned neural representations that match the statistics of the natural environment[13]. Also, the explicit engagement of brain circuits that support predictive coding in tackling complex tasks in natural environments should be studied. Feedback is critical for stateful systems, and understanding the role of feedback signals within a layer, between layers of the cortex, and between different brain areas[143,144] for a predictive model will be useful for training dynamically sparse stateful systems.

2. At the application level, dynamic sparsity could offer substantial benefits across diverse energy-constrained perception systems. Ultra-low-power intelligent sensor nodes can exploit temporal

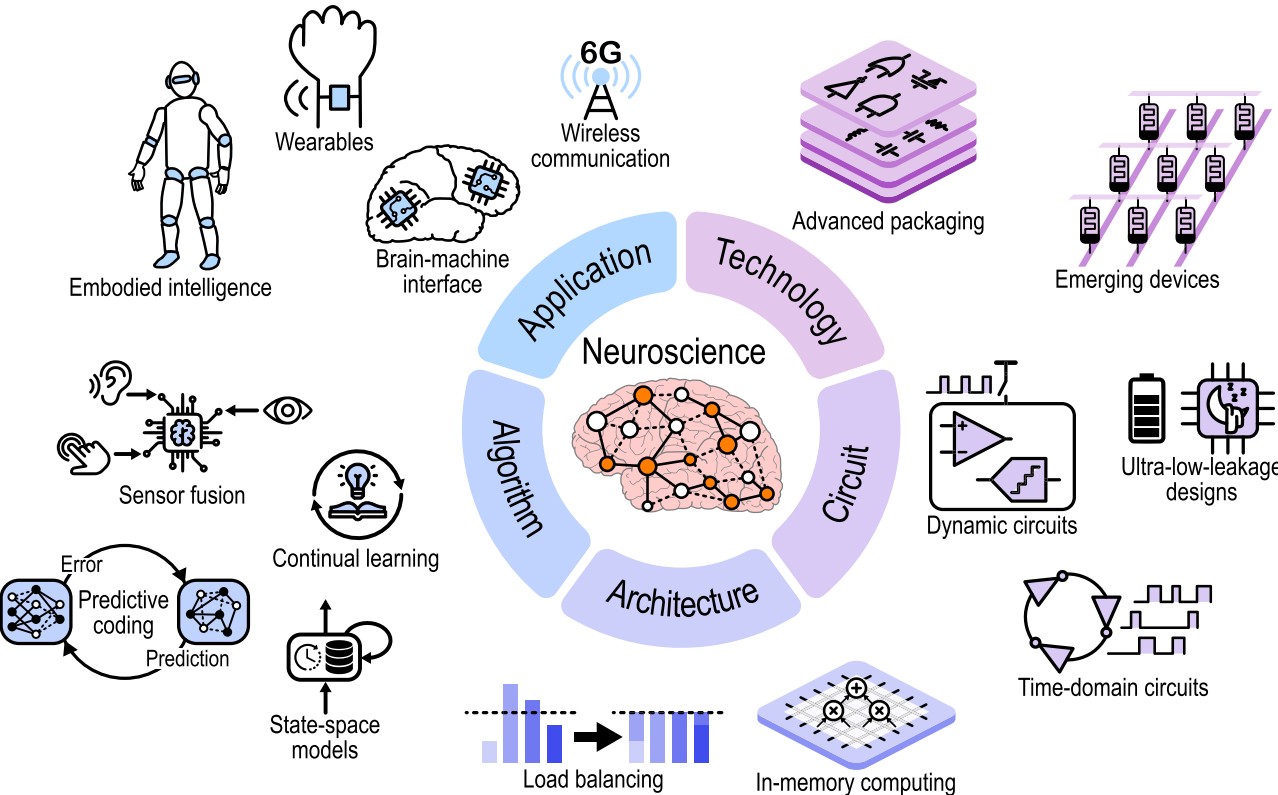

**Fig. 6 | Directions for further research.** Realizing dynamic sparsity's full potential calls for a cross-layer research effort spanning applications, algorithms, architectures, circuits, and device technologies, with neuroscience providing the foundational inspiration.

redundancy in environmental data to activate processing only when significant changes occur[145]. For implantable biomedical devices, particularly processors for neural interfaces or neuro-prostheses near or in the brain, dynamic sparsity can be extracted by leveraging the sparse nature of neural signals for reduction in computation. Wearable technologies, from hearables to multi-sensor health monitors, similarly benefit by transmitting only critical events rather than continuous data streams, simultaneously extending battery life and reducing wireless bandwidth requirements[146]. Beyond individual devices, dynamic sparsity addresses system-level challenges in future 6G wireless networks. By enabling selective data transmission and reducing redundant communication, it helps manage the power demands of inter-connecting billions of edge devices with data centers[147]. Finally, another open opportunity is to couple dynamic sparsity with continual learning to reduce resource usage while maintaining accuracy on real-world tasks[148].

3. At the algorithmic level, new sparse update schemes are needed for perception systems to efficiently process multiple dynamic data streams of different temporal scales to accomplish multiple tasks simultaneously. Innovations are also needed for future stateful systems, particularly training methods for predictive coding systems[140,141,149] that determine the predictive state at the different processing levels and the conditioning of the networks for maximal energy savings from the state-induced sparsity. We see the potential increase of dynamic sparsity without information loss by using predictive coding[23]. We also see value in investigating whether new stateful architecture, such as state-space models and RNNs, can benefit from dynamic sparsity-enhancing or exploiting methods; and whether they can act as better predictors. And yet, relatively unexplored is how dynamic sparsity could reduce the cost of continual learning in deployed systems through fewer weight updates and less memory access[150].

4. At the architectural level, we need to replace the static scheduling used in current AI accelerators with dynamic scheduling for exploiting data-dependent sparsity while limiting the resulting control overhead. Selective processing allows predictions to be determined close to the sensors, sparsifying the wake-up of more expensive modules. Dynamic schedulers need hardware support; otherwise, they will be costly. Other considerations include load balancing and efficient shared memories that allow workloads to be dynamically shifted between processing cores. We also see potential in combining dynamic sparsity with emerging architectural paradigms, such as in-memory computing. Some of the dynamic sparsity techniques are used in recent mass-produced smartphone neural processing units[111], but there are many opportunities to improve them by exploiting multiple sparsity types in combination.

5. At the circuit level, more techniques are needed to support dynamic sparsity. These include ways of using dynamic circuits and reducing the idle power of circuit blocks so that the power savings from dynamic sparsity are maximized. In addition, fine-grained dynamic sparsity exploitation could benefit from the emerging time-domain circuits[70]. Introducing gating functions coming from an auxiliary neuron or network that uses the data and states of connected neurons or other networks will help in dynamic reconfiguration or activation of subsystems. This is possible by building more efficient multiplexers, which can be rather slow (like in the brain) but need to be more energy-efficient. Low-cost storage of this configuration locally is needed, which boils down to the need for compact memory.

6. At the device technology level, the main limitation for our vision is that the data movement for the feedback mechanisms and the states are dense and ideally 3D. We need denser memories directly stacked with the compute layers. Just like the brain is a 3D interwoven structure of computing and memory, emerging

memory devices interwoven with computation and wafer stacking[68] can potentially reduce the structural dissimilarity between the brain and conventional 2D CMOS chips, enabling more faithful implementation of bio-inspired activity-driven computing[79,151]. We need to determine the area and energy cost of retaining state and moving data, and how this cost can be improved by emerging memories and advanced 3D packaging.

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

## Acknowledgements

S.Z. and S.C.L. are supported by Swiss National Science Foundation (no. 208227). C.G. is supported by Dutch Research Council (NWO) Talent Programme Veni 2023 (no. 21132) and Marie Skłodowska-Curie Actions Postdoctoral Fellowship (no. 101107534). M.V. is supported by European Research Council Seventh Framework Programme (specific programme "IDEAS", no. 101088865), the Flanders AI Research Program, and KU Leuven's Methusalem programme. Figures 3, 4, and 6 have been designed using free icon resources from flaticon.com.

## Author contributions

S.Z., C.G., T.D., M.V., and S.C.L. contributed to the conceptualization, structuring, and writing of the manuscript.

## Competing interests

The authors declare no competing interests.
