## [Transparent Peer Review file · Nature Communications]

Exploiting neuro-inspired dynamic sparsity for energy-efficient intelligent perception

Corresponding Author: Professor Shih-Chii Liu

Version 0:

Reviewer comments:

Reviewer #1

(Remarks to the Author)

The paper clearly presents a visionary stance on energy-efficient AI, proposing dynamic sparsity as a key enabler. The biological inspiration is well-argued, linking neuroscience principles such as sparse neural coding, predictive coding, and attentional mechanisms to potential AI improvements. Dynamic sparsity is definitely a promising aspect to be considered in real hardware design.

Strengths:

1. The paper draws extensively from neuroscience research to justify why dynamic sparsity is effective in biological systems.
2. Examples like predictive coding, event-driven sensing, and attention mechanisms help ground the discussion in biologically inspired principles.
3. The references to existing sparsity-aware hardware and event-based sensors are well-supported by prior work.

Weaknesses:

1. While the biological analogies are compelling, a more rigorous mathematical or computational framework for dynamic sparsity would strengthen the argument.
2. The emphasis on neuromorphic computing is interesting, but it would be useful to see a stronger comparison with traditional deep learning sparsity methods, at least discussing them in the Perspective.

Specifically, two types of dynamic sparsity in deep learning need to be discussed. (1) dynamic sparse training [1,2,3,4]; (2) mixture-of-expert [5,6]

[1] <https://arxiv.org/abs/1707.04780>

[2] <https://arxiv.org/abs/1911.11134>

[3] <https://arxiv.org/abs/2102.02887>

[4] <https://arxiv.org/abs/2106.10404>

[5] <https://www.cs.toronto.edu/~hinton/absps/Outrageously.pdf>

[6] <https://arxiv.org/abs/2006.16668>

[7] <https://arxiv.org/abs/2101.03961>

Reviewer #2

(Remarks to the Author)

The Perspective article describes a neuro-inspired approach to utilizing dynamic sparsity as a means towards more efficient and performant AI. In contrast to static sparsity, dynamic sparsity offers data-dependent opportunities to strongly reduce the degree of computation. The article describes how dynamic sparsity is utilized in the brain, provides a taxonomization and

survey of research across different computing subsystems, and finally provides an outlook of the neuro-inspired approach towards further dynamic sparsity-aware systems.

I commend the authors on the creative examples throughout their paper, namely the descriptions of a 26m² chip and the 26 hour cellular recording, as well as their thorough taxonomization and survey of dynamic sparsity technology, which contributed to an enjoyable read of the article.

Feedback:

1. The writing appears to under-represent the facets of existing work in various places.

1a. Introduction: "the spatiotemporal correlations of natural stimuli and contextual properties of real-world inputs are not yet considered". Not yet considered at all? Is this not the entire idea behind the attention mechanism of transformers?

1b. Section 1: "A stateful system also allows decisions to be made without requiring a continuous full update of the entire system. This contrasts current AI systems...". Also Section 4, Table 1: "Drastically increasing sensor modalities and data volumes is impossible in today's systems exploiting only stateless sparsity". On first read, the authors seem to be claiming that AI systems do not explore stateful sparsity at all.

1c. Section 3.4: It is worth noting that speculative decoding [1], which is common in LLM generative inference, falls into this category. In addition, this category does not fit well in the presented taxonomy of Section 2, which appears focused on the granularity of neurons or network layers. The taxonomy should be expanded or explicitly and appropriately limited.

2. The presentation of Figure 5 in Section 4 is confusing. The statement "Yet, the amount of sparsity will strongly depend on the predictive value of the state" does not make sense in context. Sub-Figure 5C needs to be referenced and explained promptly, not multiple paragraphs later in the list. As a result of this confusion, it is difficult to understand what exactly the 'neuro-inspired vision' the article is attempting to present is.

3. Section 3.2 statement "sparse activations must be compressed on the fly, thereby requiring dedicated encoder/compressor circuitry in the memory interface" is dubious. The article's reference number 78 describes a structured dynamic pruning of attention matrices, which are data-dependent like neuron activations, by using GPU kernel fusion to minimize overhead - achieving compression without dedicated circuitry.

4. Grammar and style comments

4a. Section 2: "The sparsity of all kinds"

4b. Section 3: Introductory paragraphs and Figure 4 focus around three subsections, not well aligned with the fourth subsection. Recommend to augment Figure 4 with system-module level visualization, or clarify the intra-system scope earlier.

4c. Section 3.3: Request citation for standard benchmark dataset and more context behind 26 hour audio recording.

4d. Section 3.3: Design considerations paragraph list of 3 orthogonal directions are presented awkwardly.

Overall, the article presents a strong survey and outlook on a neuro-inspired direction for dynamic sparsity-aware AI systems. My opinion is that the lack of clarity in Section 4, described in point 2, is a weakness of the article that needs to be particularly addressed.

[1] <https://pytorch.org/blog/hitchhikers-guide-speculative-decoding/>

Reviewer #3

(Remarks to the Author)

This paper offers a perspective and survey of sparsity in ANNs with specific attention towards methods that are neuro-inspired. The paper is timely given the energy costs of state-of-the-art AI algorithms. Overall, the paper is well-written and provides a nice taxonomy for dynamic sparsity. Authors should address the following comments/questions:

-The authors should better motivate the need for such a survey by comparing/contrasting with other surveys that are similar, e.g., "Dynamic Neural Networks: A Survey" by Han et al.

- "A stateful system also allows decisions to be made without requiring a continuous full update of the entire system. This contrasts current AI systems that require perfect inference accuracy for every input frame." - Please clarify these statements. What is meant by "require perfect inference accuracy for every input frame?"

- "While prior works have incorporated various forms of dynamic sparsity, they are often applied in an ad hoc and fragmented manner." Please be more specific here.

- Figure 4 does not seem to add any useful information for the reader. I recommend making the figure more detailed or removing it.

- "Sparse coding aims to represent input signals using an overcomplete set of basis vectors, ensuring that only a few

coefficients are nonzero, thereby reducing the data's dimensionality and redundancy [11]." - My understanding is that VSAs such as hyperdimensional computing usually use very high-dimensional (maybe higher than the input) encodings.

-The authors discuss the use of in-memory-computing architectures and mention that when paired with sparse activations, IMC can have reduced energy/latency. Another important aspect that should be included is the limitation/challenge of IMC designs (especially typical memory crossbar arrays) for sparse connectivity/weight sparsity. What can be done to use IMC for network layers that are not fully-connected?

-In the outlook, it may be helpful to include more discussion on how the proposed sparsity techniques can be reconciled with the sota ML architectures (e.g. transformers, deep cnns, etc.). Are these readily adapted to dynamically sparse implementation, or is a fundamental shift needed? Do the authors see these concepts being employed everywhere in the future or only for edge applications? What are the implications for AI accelerator industry?

Reviewer #4

(Remarks to the Author)

Summary:

This perspective article introduces dynamic sparsity (inspired by biological neural systems) to enhance the energy efficiency of AI systems. The authors contrast static sparsity (e.g., pruning, quantization) with dynamic sparsity, which adapts computations based on input data and context. They propose a taxonomy of dynamic sparsity across dimensions (spatial, temporal, spatiotemporal), structuredness (structured/unstructured), and statefulness (stateless/stateful). The paper discusses potential algorithmic and hardware co-design strategies for sensors, memory, and neural compute subsystems, and outlines future research directions in neuroscience, algorithms, architectures, and device technologies.

Overall, the manuscript is well written, provides a well-organized synthesis of dynamic sparsity techniques and discusses their biological underpinnings. However, its originality is constrained by the incremental nature of its contributions. The proposed taxonomy and subsystem-level analysis are valuable but build heavily on prior work in neuromorphic engineering and sparse neural networks. This work is better suited for a specialized journal rather than Nature Communications, given its incremental advances and lack of groundbreaking empirical or theoretical breakthroughs.

Strengths:

1. **Well-structured Framework:** The taxonomy of dynamic sparsity (Fig. 3) provides a structured framework for understanding and categorizing existing and future techniques. This organization clarifies the relationship between biological mechanisms (e.g., predictive coding, attention) and engineering implementations.
2. **Neuro-Biological Insights:** The emphasis on biological inspiration (e.g., sparse firing rates, predictive coding, retinal/cochlear event-based sensing) grounds the discussion in neuroscientific principles, distinguishing it from purely algorithmic approaches to sparsity.
3. **Subsystem-specific Analysis:** The exploration of dynamic sparsity across sensor, memory, and compute subsystems highlights opportunities for holistic optimization. For example, neuromorphic sensors (e.g., DVS cameras) and in-memory computing architectures are discussed as synergistic components.
4. **Forward-Looking Perspective:** The identification of six research directions (e.g., neuroscience, applications, algorithms, architectures, circuits, devices) offers a roadmap for interdisciplinary innovation. The call for hierarchical stateful systems with feedback mechanisms (Fig. 5) is interesting.

Weaknesses:

1. **Incremental Contribution:** While the taxonomy is well-structured, much of the reviewed work (e.g., event cameras, SNNs, delta networks) has been explored in prior literature. The distinction between static and dynamic sparsity is not entirely novel, as dynamic pruning and activation sparsity in neural networks (e.g., condensation of NNs, mixture-of-experts, dynamic compression, etc.) are already established. The paper's framing as a "neuro-inspired" approach does not sufficiently differentiate it from existing sparse learning paradigms.
2. **Lack of Empirical Validation:** As a perspective, the manuscript lacks experimental validation of its claims (e.g., energy savings from stateful sparsity in NNs). While speculative discussions are acceptable for this format, concrete examples or case studies (beyond citations) would strengthen the argument.
3. **Underaddressed Challenges:** The trade-offs of dynamic sparsity (e.g., control overhead, state memory footprint, latency from sparse scheduling) are mentioned but not deeply analyzed. For instance, the energy cost of maintaining stateful systems (Sec. 3.3) could negate the benefits of sparsity, yet this is only briefly acknowledged.
4. **Limited Engagement with Competing Approaches:** The manuscript does not adequately contrast dynamic sparsity with other energy-efficient AI paradigms, such as topological sparsity, hyperdimensional computing, or photonic accelerators. A comparative analysis or at least discussion would better position its contributions.

Considering the above weaknesses, I believe the manuscript in its current form is not qualified for publication in Nature Communications.

Version 1:

Reviewer comments:

Reviewer #1

(Remarks to the Author)

I thank the authors for their revisions and detailed responses to my review. After reviewing the updated manuscript and considering other reviewers' comments, I still have concerns. It appears that the authors lack a comprehensive and thorough understanding of current AI algorithms, architectures, and systems. Additionally, several statements remain overly broad or bold—such as "the spatiotemporal correlations of natural stimuli and contextual properties of real-world inputs are not yet considered"—which may mislead readers, particularly in neuroscience. Moreover, the manuscript overlooks dynamic sparse training (DST) methods, which explicitly allow not only sparsity patterns but also sparsity ratios to evolve during training. Overall, I think the review of dynamic sparsity has its merits to the neuroscience community, but I am concerned about the precision of the statements in the paper due to limited understanding of current AI advancements.

Reviewer #2

(Remarks to the Author)

Thanks to the authors for their thorough and attentive response to the review concerns. For me, all points have been addressed acceptably. In particular, Section 4, which was my main point of concern, has been extensively updated with well-drawn visuals and clarified Perspective writing. I have no further comments and recommend acceptance of the Perspective.

Reviewer #3

(Remarks to the Author)

I commend the authors on taking time to carefully review and address each of my comments. I believe that the paper is now ready for publication.

Reviewer #4

(Remarks to the Author)

The paper is well revised. I do not have further questions.

Response to Reviewers' Comments

We thank the reviewers for their thoughtful comments. Their suggestions sparked many interesting discussions among the authors, and resulted in significant updates of the manuscript. In the following, we summarize the resulting changes and address the reviewer comments point-by-point.

The major changes to the manuscript are:

1. A new title with no punctuation. It also clarifies that the focus of our Perspective is on energy-efficient *intelligent perception systems*.
2. A rewritten Introduction that emphasizes the scope of our Perspective—dynamic sparsity for energy-efficient perception—and discusses the connection with other research fields, including spiking neural networks and transformers.
3. An updated Section 1 with a more accurate description of stateful computation in biological and artificial neural networks.
4. Modified Figure 3A and 3B for better contrasting of dynamic sparsity with static sparsity. They highlight the data-dependent, sparse processing flow of dynamic sparsity during inference.
5. A new Box 2 that provides a formal definition and taxonomy of dynamic sparsity.
6. Additional paragraphs at the start of Section 2 that address the differences and connections between our treatment of dynamic sparsity and prior works on Dynamic Sparse Training, Mixture-of-Expert, speculative decoding, and dynamic neural networks.
7. A more elaborate Figure 4 that illustrates the dynamic sparsity enhancing and exploitation techniques discussed in Section 3. Figure 4A, 4B, and 4C detail how dynamic sparsity can be applied in the sensor, memory, and neural-compute subsystem of an intelligent perception system. Figure 4D is added to provide a visual example of the system-level dynamic sparsity covered in Section 3.4.
8. A clearer description of our perspective at the start of Section 4 along with a an updated Figure 5 that explains the vision for dynamic sparsity that go beyond the techniques described in Section 3.
9. A new Figure 6 that illustrates the research directions for neuro-inspired dynamic sparsity, complementing the list of descriptions at the end of Section 4.

1 Reviewer #1 (Remarks to the Author):

The paper clearly presents a visionary stance on energy-efficient AI, proposing dynamic sparsity as a key enabler. The biological inspiration is well-argued, linking neuroscience principles such as sparse neural coding, predictive coding, and attentional mechanisms to potential AI improvements. Dynamic sparsity is definitely a promising aspect to be considered in real hardware design.

Strengths:

1. The paper draws extensively from neuroscience research to justify why dynamic sparsity is effective in biological systems.
2. Examples like predictive coding, event-driven sensing, and attention mechanisms help ground the discussion in biologically inspired principles.
3. The references to existing sparsity-aware hardware and event-based sensors are well-supported by prior work.

Response: Thank you for the positive feedback!

Weaknesses:

1. While the biological analogies are compelling, a more rigorous mathematical or computational framework for dynamic sparsity would strengthen the argument.

Response: Thank you for bringing this issue to our attention. To clarify the definition of dynamic sparsity and provide a rigorous framework for future research in this direction, we have added **Box 2** to the revised manuscript.

2. The emphasis on neuromorphic computing is interesting, but it would be useful to see a stronger comparison with traditional deep learning sparsity methods, at least discussing them in the Perspective. Specifically, two types of dynamic sparsity in deep learning need to be discussed. (1) Dynamic Sparse Training [1,2,3,4]; (2) Mixture-of-Expert [5,6,7]

[1] <https://arxiv.org/abs/1707.04780>

[2] <https://arxiv.org/abs/1911.11134>

[3] <https://arxiv.org/abs/2102.02887>

[4] <https://arxiv.org/abs/2106.10404>

[5] <https://www.cs.toronto.edu/~hinton/absps/Outrageously.pdf>

[6] <https://arxiv.org/abs/2006.16668>

[7] <https://arxiv.org/abs/2101.03961>

Response: Thank you for pointing out these related areas of research, and we have included a discussion of these approaches in **Section 2**.

In Mixture of Experts (MoE) [5-7], a small subset of the specialized sub-networks (“experts”) are selectively activated depending on the input data. Therefore, MoE is an example of coarse-grained, network-level dynamic sparsity, showcasing that this neuro-inspired principle can be applied to state-of-the-art deep networks.

In Dynamic Sparse Training (DST) [1-4], the sparse connectivity of the network is dynamically adjusted during training but fixed during inference. Therefore, we believe dynamic sparsity, which focuses on context-dependent sparse processing for efficient inference, is distinct from DST. Nevertheless, dynamic sparsity could potentially complement DST to further reduce the training cost, especially for continual learning, which is mentioned in **Section 4**.

2 Reviewer #2 (Remarks to the Author):

The Perspective article describes a neuro-inspired approach to utilizing dynamic sparsity as a means towards more efficient and performant AI. In contrast to static sparsity, dynamic sparsity offers data-dependent opportunities to strongly reduce the degree of computation. The article describes how dynamic sparsity is utilized in the brain, provides a taxonomization and survey of research across different computing subsystems, and finally provides an outlook of the neuro-inspired approach towards further dynamic sparsity-aware systems.

I commend the authors on the creative examples throughout their paper, namely the descriptions of a 26 m² chip and the 26-hour cellular recording, as well as their thorough taxonomization and survey of dynamic sparsity technology, which contributed to an enjoyable read of the article.

Feedback:

1. The writing appears to under-represent the facets of existing work in various places.

Response: Thank you for pointing us to many more synergies between our taxonomy and recent works in the area of transformers and generative AI. We did not include these models in our first submission because we had focused on inference techniques for perception (sensory) systems at the edge rather than cloud-scale computing. After more internal discussions, sparked by your interesting comments, we believe that we need to **a)** make our scope clearer in the manuscript; and **b)** also include several of the techniques mentioned by you, as some of these techniques are indeed also applicable to edge perception systems.

We next outline in more detail the changes to the manuscript based on your suggestions and comments.

- 1a. Introduction: “the spatiotemporal correlations of natural stimuli and contextual properties of real-world

inputs are not yet considered”. Not yet considered at all? Is this not the entire idea behind the attention mechanism of transformers?

Response: You are right that transformers dynamically attend to the tokens in a data-driven (and hence context-aware) manner. We now added a statement (with relevant references) acknowledging this in our **Introduction**. Yet, we also comment that while the attention mechanism already takes into account the contextual information of the token sequence, which could be video, audio or texts, it is typically performed in a *dense* fashion: The goal of the attention mechanism is primarily towards increased accuracy rather than reduced computational cost. Therefore, while the attention mechanism is effective and widely used, there is still a lot of margin for further exploiting dynamic sparsity in a data-driven and context-aware fashion, as nature does.

We further discuss several techniques for reducing the inference cost of transformers in **Section 2** and—more extensively—in **Section 3.4**, including Mixture of Experts (MoE) and speculative decoding. MoE takes the data-driven attention mechanism one step further and dynamically activates only parts of a transformer model depending on the incoming token stream, thereby saving computational cost. Speculative decoding, on the other hand, uses a lightweight draft model to propose tokens that will be verified by a full model, allowing increased parallelism and hence execution efficiency for the full model (see also our response to your comment **1c**). These and many other emerging forms of data-driven dynamic sparsity are what we hope to cover with our introduced taxonomy.

1b. Section 1: “A stateful system also allows decisions to be made without requiring a continuous full update of the entire system. This contrasts current AI systems...”. Also Section 4, Table 1: “Drastically increasing sensor modalities and data volumes is impossible in today’s systems exploiting only stateless sparsity”. On first read, the authors seem to be claiming that AI systems do not explore stateful sparsity at all.

Response: Thank you for pointing this out. You are completely right that the tone of the original manuscript suggested that stateful sparsity is not explored at all yet. We have now updated various sections of the manuscript to provide a more nuanced view, with the aim to bring across that *stateful sparsity has been exploited but in a rather fragmented way, and mainly with coarse-grained methods*. With our Perspective and the proposed taxonomy, we hope to make readers aware of the complete landscape of dynamic sparsity, and highlight the research opportunities that will further push the frontiers of stateful dynamic sparsity.

Modifications have been made, among others, in the **Introduction** (to set the scope and goal of the Perspective), **Section 1** (to discuss the use of states in current neural network architectures), and **Section 4** (to clarify the role of states in the dynamically-sparse perception systems that we envision). In Section 4, we also updated the left column of **Table 1** to bring more nuance. Finally, we put a stronger focus on our taxonomy with the newly added **Box 2**, which helps to position and organize the many state-of-the-art techniques emerging in this field.

1c. Section 3.4: It is worth noting that speculative decoding [1], which is common in LLM generative inference, falls into this category. In addition, this category does not fit well in the presented taxonomy of Section 2, which appears focused on the granularity of neurons or network layers. The taxonomy should be expanded or explicitly and appropriately limited.

[1] <https://pytorch.org/blog/hitchhikers-guide-speculative-decoding/>

Response: It is a very interesting suggestion to link our discussions on hierarchical exploitation of dynamic sparsity to speculative decoding, and we have updated **Section 3.4** to address their connection. In our view, they are orthogonal to a certain extent: Speculative decoding always runs *both* the lightweight draft model as well as the heavyweight target model on every token, and hence does not reduce computations. Instead, the technique improves inference efficiency by allowing the target model to run in batch mode, which is more efficient compared to executing it token by token. As such, the execution efficiency is context-dependent in a data-driven way. Yet, it does not reduce the total number of computations—even more computes are required, especially if a token is rejected by the heavyweight model. In contrast, the discussed hierarchical dynamic sparsity focuses on using states to shut down certain downstream or upstream hardware blocks to save compute. Nevertheless, there are certainly relevant links between these

methods, and one can only expect the two to converge more in the future. For example, the speculative decoding techniques could start to use more aggressive dynamic gating techniques.

2. The presentation of Figure 5 in Section 4 is confusing. The statement “Yet, the amount of sparsity will strongly depend on the predictive value of the state” does not make sense in context. Sub-Figure 5C needs to be referenced and explained promptly, not multiple paragraphs later in the list. As a result of this confusion, it is difficult to understand what exactly the ‘neuro-inspired vision’ the article is attempting to present is.

Response: Thank you for your comment. We have drastically updated **Figure 5** to better illustrate the dynamic sparsity differences between stateless and stateful systems. We also highlight the neural inspiration from both the connectivity of the brain areas in the visual pathway and the possibility of neuro-inspired models such as predictive coding that benefit from stateful networks.

3. Section 3.2 statement “sparse activations must be compressed on the fly, thereby requiring dedicated encoder/compressor circuitry in the memory interface” is dubious. The article’s reference number [78] describes a structured dynamic pruning of attention matrices, which are data-dependent like neuron activations, by using GPU kernel fusion to minimize overhead—achieving compression without dedicated circuitry.

Response: Thank you very much for pointing this out. We very much agree the previous statement was not very clear. We now updated the **Design considerations** paragraph of **Section 3.2** to include a more detailed discussion of the overhead of making use of dynamic sparsity in activations. It is definitely possible to encode/compress the sparse activations on the fly without dedicated circuitry, but that leads to higher overhead in terms of latency and throughput if it is a purely GPU software-based solution. In applications where latency and throughput are critical, dedicated hardware solutions are typically used to minimize the overhead in compressing/encoding the sparse activations so that the meaningful values (e.g. the non-zero values) are dispatched to their corresponding compute units more efficiently.

4. Grammar and style comments

4a. Section 2: “The sparsity of all kinds”.

Response: Thanks for the comment. We updated the first sentence of **Section 2** in the manuscript to make it smoother: “Sparsity plays a crucial role in both biological and artificial perception system.”

4b. Section 3: Introductory paragraphs and Figure 4 focus around three subsections, not well aligned with the fourth subsection. Recommend to augment Figure 4 with system-module level visualization, or clarify the intra-system scope earlier.

Response: Thank you for pointing out the misalignment between Figure 4 and Section 3 (in particular, Section 3.4). We modified the **introductory paragraph of Section 3** to also cover system-level dynamic sparsity, which is then further elaborated in Section 3.4. Additionally, we redraw **Figure 4** to better visualize the dynamic sparsity techniques discussed in Section 3, and every subsection is illustrated with a dedicated subfigure. In particular, **Figure 4D** is added for the system-level dynamic sparsity discussed in Section 3.4.

4c. Section 3.3: Request citation for standard benchmark dataset and more context behind 26 hour audio recording.

Response: Thanks for pointing out the brevity of this description. We have now expanded and moved the description to the beginning of **Section 4** to clarify the results we observed over two separate recordings, each spanning more than 24h.

We believe that standard benchmarks underestimated the potential benefits of dynamic sparsity. To illustrate this, we recorded on two separate days using a personal audio cellphone from one of the authors during their normal daily use, covering public transportation travel, time in an office, and sleeping hours. We checked the effective sparsity using a previously reported spoken language understanding (SLU) network [2].

Fig. 1 shows the effective dynamic sparsity averaged per hour for the input audio features (Δ_{feat}) and the two hidden layers of the GRU RNN (Δ_{H1} and Δ_{H2}). We observed that the input feature (frequency band

amplitude) sparsity is typically around 99%. The lowest sparsity is in the first hidden layer, but even this has a minimum hourly average of about 95%. This sparsity exceeded our expectation significantly, and so we consider it worth reporting in this Perspective since it shows that dynamic sparsity can reduce compute and memory access by more than $20\times$ compared with standard dense inference using datasets.

Figure 1: Measured dynamic sparsity in a SLU RNN from one of the two cellphone recordings in normal working days of one of the authors. This RNN has 16 input features and two GRU layers each with 64 hidden units. The delta threshold used here was 0.125, the same as reported in [2]. Note that the sparsity level on the y-axis span from 0.96 to 1.

[2] S. Zhou *et al.*, “An 8.62 μ W 75 dB-DR_{SoC} End-to-End Spoken-Language-Understanding SoC With Channel-Level AGC and Temporal-Sparsity-Aware Streaming-Mode RNN,” in *IEEE International Solid-State Circuits Conference*, pp. 238-240, 2025.

4d. Section 3.3: Design considerations paragraph list of 3 orthogonal directions are presented awkwardly.

Response: Thank you for pointing out the lack of clarity in this paragraph. We improved the **Design considerations** paragraph of **Section 3.3** by separating it into two paragraphs. The first one addresses the control and scheduling overhead of dynamic sparsity, while the second one analyzes the memory overhead due to states. In both of the new paragraphs, we provide concrete design examples to explain the root cause of the overhead and the possible mitigating methods.

Overall, the article presents a strong survey and outlook on a neuro-inspired direction for dynamic sparsity-aware AI systems. My opinion is that the lack of clarity in Section 4, described in point 2, is a weakness of the article that needs to be particularly addressed.

Response: Thank you for the comments. We have now rewritten the first part of **Section 4**, redrawn **Figure 5** and provided a new **Figure 6** to clarify the outlook for our neuro-inspired vision of future dynamic sparsity-aware perception systems. We also updated item 2 of the research directions list to discuss specific examples on the topics covered by Figure 6 as we are not sure whether the “point 2” is pointing to the second comment or item 2 of the research directions list in Section 4.

3 Reviewer #3 (Remarks to the Author):

This paper offers a perspective and survey of sparsity in ANNs with specific attention towards methods that are neuro-inspired. The paper is timely given the energy costs of state-of-the-art AI algorithms. Overall, the paper is well-written and provides a nice taxonomy for dynamic sparsity. Authors should address the following comments/questions.

1. The authors should better motivate the need for such a survey by comparing/contrasting with other surveys that are similar, e.g., “Dynamic Neural Networks: A Survey” by Han et al.

Response: Thank you for pointing us to the survey on dynamic neural networks by Han et al [1], which is now cited (among other related works) in the revised manuscript as reference [49].

Indeed, our Perspective on neuro-inspired dynamic sparsity is related to many excellent surveys on topics such as dynamic neural networks and neural network sparsity. These papers summarized the *algorithmic aspect* of dynamic sparsity *within neural networks*. However, we emphasize that our focus is on energy-efficient *intelligent perception systems*. Therefore, as explained in the **introductory paragraphs** of **Section 2**, our Perspective is different from prior surveys because it presents a systematic treatment of dynamic sparsity *throughout the entire perception pipeline*—from sensory periphery to higher-level decision-making—and encompasses both the *algorithmic and hardware aspects*. We also modified the **manuscript title** as well as the **Introduction** of the paper to clarify our scope.

[1] Y. Han *et al.*, “Dynamic Neural Networks: A Survey,” in *IEEE Transactions on Pattern Analysis and Machine Intelligence*, vol. 44, no. 11, pp. 7436-7456, 1 Nov. 2022.

2. “A stateful system also allows decisions to be made without requiring a continuous full update of the entire system. This contrasts current AI systems that require perfect inference accuracy for every input frame.” — Please clarify these statements. What is meant by “require perfect inference accuracy for every input frame?”

Response: Thank you for pointing out the lack of clarity of this sentence. We now see that the sentence is incorrect, and we have modified the entire paragraph (i.e., the third paragraph) accordingly in **Section 1**.

3. “While prior works have incorporated various forms of dynamic sparsity, they are often applied in an ad hoc and fragmented manner.” Please be more specific here.

Response: Thank you for the comment. We have modified this sentence in the revised manuscript to the following: “Prior works that have discussed and incorporated various forms of dynamic sparsity are often applied to solve specific, isolated problems, resulting in a fragmented landscape.” In addition, we added more examples of prior works as part of a new paragraph.

4. Figure 4 does not seem to add any useful information for the reader. I recommend making the figure more detailed or removing it.

Response: Thank you for the suggestion. In the revised manuscript, we included a more elaborate **Figure 4** to visualize the dynamic sparsity enhancing and exploitation techniques presented in Section 3. The three subsystems (sensor, memory, and neural-compute) shown as abstract blocks in the original figure are now illustrated by three dedicated subfigures (Figure 4A, 4B, and 4C). In addition, a new subfigure Figure 4D is added to visualize system-level dynamic sparsity (Section 3.4).

5. “Sparse coding aims to represent input signals using an overcomplete set of basis vectors, ensuring that only a few coefficients are nonzero, thereby reducing the data’s dimensionality and redundancy [11].” — My understanding is that VSAs such as hyperdimensional computing usually use very high-dimensional (maybe higher than the input) encodings.

Response: You are correct that VSAs (also known as hyperdimensional computing) use very high-dimensional vectors to encode information, and that the dimension of the encoding vector can be higher than that of the input vector. Since the original statement in **Section 3.1** could be misleading, we rephrased it to be ‘the data representation is highly sparse’.

6. The authors discuss the use of in-memory-computing architectures and mention that when paired with sparse activations, IMC can have reduced energy/latency. Another important aspect that should be included is the limitation/challenge of IMC designs (especially typical memory crossbar arrays) for sparse connectivity/weight sparsity. What can be done to use IMC for network layers that are not fully-connected?

Response: Thank you for bringing up this aspect of sparsity in IMC designs. We agree that incorporating *weight sparsity* has led to many interesting research avenues over the last decade. However, the focus of our Perspective is on exploiting dynamic *activation sparsity*. In addition, elaborating on all different techniques related to weight sparsity in the different subsections of Section 3 would, in our opinion, dilute the focus and the message of the article. This is why we decided to only comment on state-of-the-art techniques for exploiting activation sparsity.

To avoid similar confusion with the future readers of the article, we made several improvements in

the manuscript to better delineate the focus of the article, including, among other places, the fourth paragraph of the **Introduction**. Moreover, in **Section 3.2**, we also briefly comment on the possibility to exploit weight sparsity in IMC designs (reference [90] of the revised manuscript), without going into further detail.

7. In the outlook, it may be helpful to include more discussion on how the proposed sparsity techniques can be reconciled with the sota ML architectures (e.g. transformers, deep CNNs, etc.). Are these readily adapted to dynamically sparse implementation, or is a fundamental shift needed? Do the authors see these concepts being employed everywhere in the future or only for edge applications? What are the implications for AI accelerator industry?

Response:

Thank you for the insightful suggestion.

In the revised Section 4, we now explicitly discuss the relevance and applicability of dynamic sparsity to state-of-the-art architectures such as Transformers and CNNs. Specifically, we explain that stateless dynamic sparsity has already been adopted in both academia and industry, e.g., through Mixture-of-Experts (MoE) in large language models and zero-skipping in ReLU-based CNNs. These approaches map naturally to hardware-level optimizations like zero-gating and skipping, and are already supported by commercially available AI accelerators, several of whom are cited in the revised text.

However, we argue that future scalability and energy efficiency gains will increasingly require stateful dynamic sparsity, which more closely mimics the computational strategies of biological systems (Section 1). Stateful approaches—such as delta networks for RNNs and temporally gated CNNs/Transformers—enable the reuse of internal memory to avoid redundant computation. While more challenging to integrate into current hardware-software stacks, they promise higher sparsity levels, better context-awareness, and thus larger efficiency gains.

Regarding the scope of applicability: although our examples and system-level arguments focus on edge intelligence and perception systems, the underlying principles generalize to other domains. Section 3.4 highlights recent large-scale models using MoE and speculative decoding in the cloud, where modular dynamic sparsity is already emerging. We, however, chose not to expand this direction further in the outlook to retain a clear focus on intelligent perception systems and edge systems, as now made explicit in the revised title and Introduction.

Finally, the implications for the AI accelerator industry are substantial. As outlined in items 4 to 6 of the Research Directions, dynamic sparsity—especially stateful and structured variants—demands architectural shifts in scheduling, memory hierarchies, gating mechanisms, and 3D device integration. These are not incremental changes but signal a transition toward more neuromorphic-inspired, context-aware hardware-software co-design, essential to sustaining energy-efficient AI in both edge and datacenter deployments.

4 Reviewer #4 (Remarks to the Author):

Summary: This perspective article introduces dynamic sparsity (inspired by biological neural systems) to enhance the energy efficiency of AI systems. The authors contrast static sparsity (e.g., pruning, quantization) with dynamic sparsity, which adapts computations based on input data and context. They propose a taxonomy of dynamic sparsity across dimensions (spatial, temporal, spatiotemporal), structuredness (structured/unstructured), and statefulness (stateless/stateful). The paper discusses potential algorithmic and hardware co-design strategies for sensors, memory, and neural compute subsystems, and outlines future research directions in neuroscience, algorithms, architectures, and device technologies.

Overall, the manuscript is well written, provides a well-organized synthesis of dynamic sparsity techniques and discusses their biological underpinnings. However, its originality is constrained by the incremental nature of its contributions. The proposed taxonomy and subsystem-level analysis are valuable but build

heavily on prior work in neuromorphic engineering and sparse neural networks. This work is better suited for a specialized journal rather than Nature Communications, given its incremental advances and lack of groundbreaking empirical or theoretical breakthroughs.

Strengths:

- 1. Well-structured Framework:** The taxonomy of dynamic sparsity (Fig. 3) provides a structured framework for understanding and categorizing existing and future techniques. This organization clarifies the relationship between biological mechanisms (e.g., predictive coding, attention) and engineering implementations.
- 2. Neuro-Biological Insights:** The emphasis on biological inspiration (e.g., sparse firing rates, predictive coding, retinal/cochlear event-based sensing) grounds the discussion in neuroscientific principles, distinguishing it from purely algorithmic approaches to sparsity.
- 3. Subsystem-specific Analysis:** The exploration of dynamic sparsity across sensor, memory, and compute subsystems highlights opportunities for holistic optimization. For example, neuromorphic sensors (e.g., DVS cameras) and in-memory computing architectures are discussed as synergistic components.
- 4. Forward-Looking Perspective:** The identification of six research directions (e.g., neuroscience, applications, algorithms, architectures, circuits, devices) offers a roadmap for interdisciplinary innovation. The call for hierarchical stateful systems with feedback mechanisms (Fig. 5) is interesting.

Response: Thank you for the positive feedback!

Weaknesses:

- 1. Incremental Contribution:** While the taxonomy is well-structured, much of the reviewed work (e.g., event cameras, SNNs, delta networks) has been explored in prior literature. The distinction between static and dynamic sparsity is not entirely novel, as dynamic pruning and activation sparsity in neural networks (e.g., condensation of NNs, mixture-of-experts, dynamic compression, etc.) are already established. The paper’s framing as a “neuro-inspired” approach does not sufficiently differentiate it from existing sparse learning paradigms.

Response: Thank you for the comment. We first emphasize that this submission is a perspective, and therefore, the contribution of novel results is not to be expected.

Similar to our response to Comment 1 of Reviewer 3, we agree that our Perspective on neuro-inspired dynamic sparsity is related to many excellent surveys on topics such as dynamic neural networks and neural network sparsity. These papers summarized the *algorithmic aspect* of dynamic sparsity *within neural networks*. However, we emphasize that our focus is on energy-efficient *intelligent perception systems*. As explained in the **introductory paragraphs** of **Section 2**, our Perspective is different from prior surveys because it presents a systematic treatment of dynamic sparsity *throughout the entire perception pipeline*—from sensory periphery to higher-level decision-making—and encompasses both the *algorithmic and hardware aspects*. Our Perspective further contrasts the two sparsity types for stateless and stateful networks, and makes an argument for further exploration of stateful networks for intelligent perception systems. In the Outlook section, we discuss how two neuro-inspired mechanisms, attention and predictive coding can be enabled through stateful networks.

- 2. Lack of Empirical Validation:** As a perspective, the manuscript lacks experimental validation of its claims (e.g., energy savings from stateful sparsity in NNs). While speculative discussions are acceptable for this format, concrete examples or case studies (beyond citations) would strengthen the argument.

Response: Thank you for the suggestion. Because this submission is a Perspective, we did not present *new* experimental results in the manuscript. Instead, we aimed at analyzing existing literature through a unified framework while proposing directions for future research. To strengthen our argument, we have discussed several design examples incorporating dynamic sparsity in **Section 3**. In the revised manuscript, we also added a more quantitative case study of the system design trade-offs when using dynamic sparsity, as explained in our response to the next comment (Comment 3).

- 3. Underaddressed Challenges:** The trade-offs of dynamic sparsity (e.g., control overhead, state memory footprint, latency from sparse scheduling) are mentioned but not deeply analyzed. For instance, the energy

cost of maintaining stateful systems (Sec. 3.3) could negate the benefits of sparsity, yet this is only briefly acknowledged.

Response: Thank you for your comments! Indeed, we missed some quantitative analysis of the trade-offs of dynamic sparsity. We have extended the **Design considerations** paragraph of **Section 3.3** to two paragraphs which detail the trade-offs of dynamic sparsity.

4. Limited Engagement with Competing Approaches: The manuscript does not adequately contrast dynamic sparsity with other energy-efficient AI paradigms, such as topological sparsity, hyperdimensional computing, or photonic accelerators. A comparative analysis or at least discussion would better position its contributions.

Response: Thank you for suggesting the potential connection between dynamic sparsity and other energy-efficient AI paradigms. Although a thorough comparative analysis is beyond the scope of our Perspective, we address the three paradigms mentioned in the comments here.

Topological sparsity In a recent work [1] on topological sparse training, the authors combined Dynamic Sparse Training (DST) with structured pruning to reduce the parameter size of large language models. As already explained in **Section 2** of the revised manuscript and our response to Comment 2 of Reviewer 1, the sparsity connectivity of the resulting network is still static and input-agnostic at the inference stage, while for dynamic sparsity, the sparse processing flow is dynamically adjusted depending on the system inputs and states.

Hyperdimensional computing To the best of our knowledge, hyperdimensional computing is a term used interchangeably with vector symbolic architecture (VSA), which has been presented in the **Stateless methods** paragraph of **Section 3.1**. In the revised manuscript, we added a comment that VSA is also known as hyperdimensional computing to avoid confusion.

Photonic accelerators In our view, dynamic sparsity and photonic accelerators are orthogonal paradigms towards energy-efficient neural computing. Photonic accelerators could potentially benefit from dynamic sparsity by—for example—selectively activating the high-power components (e.g., lasers and transimpedance amplifiers) depending on the characteristics of the input data.

- [1] A. Dhurandhar *et al.*, “NeuroPrune: A Neuro-inspired Topological Sparse Training Algorithm for Large Language Models,” in *Findings of the Association for Computational Linguistics*, pp. 2416-2430, Aug. 2024.
- [2] B.J. Shastri *et al.*, “Photonics for Artificial Intelligence and Neuromorphic Computing,” *Nature Photonics*, vol. 15, no. 2, pp. 102-114, 2021.

Considering the above weaknesses, I believe the manuscript in its current form is not qualified for publication in Nature Communications.

Response: We have made extensive changes to various sections of the manuscript following constructive suggestions of the reviewers. We hope that this new version which clarifies the goal of the Perspective will show its value.

Response to Reviewers' Comments

Exploiting neuro-inspired dynamic sparsity for energy-efficient intelligent perception

1 Reviewer #1 (Remarks to the Author):

I thank the authors for their revisions and detailed responses to my review. After reviewing the updated manuscript and considering other reviewers' comments, I still have concerns. It appears that the authors lack a comprehensive and thorough understanding of current AI algorithms, architectures, and systems. Additionally, several statements remain overly broad or bold—such as “the spatiotemporal correlations of natural stimuli and contextual properties of real-world inputs are not yet considered”—which may mislead readers, particularly in neuroscience. Moreover, the manuscript overlooks dynamic sparse training (DST) methods, which explicitly allow not only sparsity patterns but also sparsity ratios to evolve during training. Overall, I think the review of dynamic sparsity has its merits to the neuroscience community, but I am concerned about the precision of the statements in the paper due to limited understanding of current AI advancements.

Response: We thank the reviewer for evaluating the revised manuscript and for providing additional comments.

To address earlier feedback of other reviewers, we had already added in the revised manuscript a discussion of dynamic sparse training (DST) in the section “Types of Dynamic Sparsity” and refined the description of context-aware processing in the “Introduction”. We also note that the quoted sentence on spatiotemporal correlations was not in the revised manuscript.

Regarding the statement that “the authors lack a comprehensive and thorough understanding of current AI algorithms, architectures, and systems,” we feel this comment is too broad and unspecific to guide further revisions. In the absence of actionable points, we have not added additional material in response.

2 Reviewer #2 (Remarks to the Author):

Thanks to the authors for their thorough and attentive response to the review concerns. For me, all points have been addressed acceptably. In particular, Section 4, which was my main point of concern, has been extensively updated with well-drawn visuals and clarified Perspective writing. I have no further comments and recommend acceptance of the Perspective.

Response: Thank you for the positive feedback and your contribution in improving the manuscript.

3 Reviewer #3 (Remarks to the Author):

I commend the authors on taking time to carefully review and address each of my comments. I believe that the paper is now ready for publication.

Response: Thank you for the positive feedback and your contribution in improving the manuscript.

4 Reviewer #4 (Remarks to the Author):

The paper is well revised. I do not have further questions.

Response: Thank you for the positive feedback and your contribution in improving the manuscript.